# DistillHGNN: A Knowledge Distillation Approach for High-Speed Hypergraph Neural Networks

**Saman Forouzandeh, Parham Moradi & Mahdi Jalili**
School of Engineering
RMIT University
Melbourne, Australia
`{saman.forouzandeh,parham.moradi,mahdi.jalili}@rmit.edu.au`

## Abstract

This paper introduces a novel framework designed to significantly enhance the inference speed and memory efficiency of Hypergraph Neural Networks (HGNNs) while maintaining their high accuracy. Our approach, named DistillHGNN, employs an advanced teacher-student knowledge distillation strategy, where the teacher model comprises an HGNN and a Multi-Layer Perceptron (MLP). In this setup, the HGNN generates embeddings, which the MLP subsequently processes to predict soft labels. The student model consists of a lightweight Graph Convolutional Network (GCN), TinyGCN, paired with an MLP and optimised for online prediction. We leverage contrastive learning to train both TinyGCN and HGNN simultaneously, facilitating the transfer of high-order and structural knowledge from the HGNN to the TinyGCN. Additionally, the teacher employs a mechanism to transfer knowledge to the student model through soft labels. This dual transfer mechanism enables the student to effectively capture complex dependencies while benefiting from a lightweight GCN's faster inference and lower computational cost. The student is trained using both labelled data and soft labels provided by the teacher, with contrastive learning further ensuring that the student retains high-order relationships. This makes the proposed method efficient and suitable for real-time applications, achieving performance comparable to traditional HGNNs but with significantly reduced resource requirements. Experimental results on several real-world datasets demonstrate that our method significantly reduces inference time while maintaining accuracy comparable to HGNN, and it achieves higher accuracy than state-of-the-art techniques, like LightHGNN, with a similar inference time.

## 1 Introduction

Hypergraphs, with their ability to capture multi-node relationships through degree-free hyperedges, offer a significant advantage over traditional graphs in modeling complex high-order interactions (Fan et al., 2021). As a result, several Hypergraph Neural Networks (HGNNs) have been developed to tackle various tasks, such as node classification in citation networks, recommendation in bipartite graphs, and link prediction in biological networks (Zeng et al., 2024). Despite these advances, the widespread adoption of HGNNs in large-scale industrial applications still needs to be improved. This is primarily due to the heavy reliance on hypergraph structures during inference, which requires substantial memory and computational resources (Yu et al., 2024). As the hypergraph size and the HGNNs' depth increase, the inference time and memory requirements grow exponentially, posing significant challenges for their deployment in real-world scenarios where speed and efficiency are crucial (Feng et al., 2024). This disparity highlights the need for more lightweight and scalable solutions that can harness the representational power of HGNNs while being suitable for high-speed, resource-constrained environments.

The dependency of HGNNs on the hypergraph structure arises from their message-passing mechanism, which involves complex high-order interactions between vertices and hyperedges. To address

this challenge, recent methods like Graph-Less Neural Networks (GLNN) (Zhang et al., 2022), and Noise-robust Structure-aware MLPs On Graphs (NOSMOG) (Tian et al., 2022) and Knowledge-inspired Reliable Distillation (KRD)(Wu et al., 2023) have aimed to eliminate graph dependencies by distilling knowledge from GNNs to MLPs. However, these methods focus on simple graph structures, using soft labels or pairwise edge information as supervision, and are insufficient for hypergraphs, where hyperedges connect multiple vertices, leading to more intricate neighborhoods. Consequently, MLPs, despite their scalability and graph independence, often underperform on hypergraph data, showing an average decline in accuracy compared to HGNNs. Recently, LightHGNN has extended the GLNN framework to hypergraphs by distilling knowledge from HGNNs into MLPs using soft labels. While this method effectively transfers class information, it cannot convey the complex high-order relationships and structural knowledge inherent in hypergraphs. Soft labels alone are insufficient to capture these intricate dependencies. This limitation raises an important question: *Can we develop a strategy that distills soft labels and transfer the hypergraph's structural and high-order knowledge to the student model, ensuring a more comprehensive knowledge transfer*?

**Present work**. In this paper, we present DistillHGNN, a novel framework that significantly improves the inference speed and memory efficiency of HGNNs while maintaining high accuracy. DistillHGNN leverages a comprehensive teacher-student knowledge distillation approach. The teacher model includes an HGNN and an MLP, where the HGNN captures complex relationships within the hypergraph and generates node embeddings. These embeddings are then passed to the MLP, which produces soft labels. Together, these embeddings and soft labels form the knowledge to be transferred to the student model. The student model includes a lightweight GCN called TinyGCN and an MLP. TinyGCN, designed for efficient learning, consists of a single-layer GCN without non-linear activation functions to reduce computational complexity. A contrastive learning strategy maximises the similarity between the embeddings generated by the HGNN and TinyGCN, effectively transferring high-order structural knowledge to the student model. This strategy enables TinyGCN to replicate the behaviour of the HGNN while operating at a significantly lower computational cost. Moreover, soft labels are transferred as supplementary labelled data, further using the training process of the student model. This dual transfer mechanism allows the TinyGCN and MLP in the student model to inherit both high-order dependencies and structural knowledge from the HGNN, resulting in faster inference and improved performance. Novelties of the Proposed Method are:

1. DistillHGNN transfers soft labels and structural knowledge from HGNN to TinyGCN, resulting in richer and more effective knowledge distillation than methods like LightHGNN.

2. DistillHGNN utilises a contrastive learning strategy to maximise the similarity between embeddings generated by the HGNN and TinyGCN, effectively transferring high-order structural knowledge to the student model.

3. TinyGCN is streamlined to a single layer without activation functions, reducing computational complexity while effectively capturing the high-order relationships of the teacher HGNN.

4. The proposed method achieves inference speeds comparable to LightHGNN(Feng et al., 2024) but with higher accuracy, making it highly suitable for real-time and large-scale applications.

5. By integrating the representational power of HGNNs with the lightweight nature of TinyGCN, DistillHGNN delivers fast and accurate predictions while maintaining low memory and computational requirements.

## 2 PRELIMINARIES

**Graph and Hypergraph:** A graph $G = (V, E)$ consists of a set of nodes $V$ and edges $E$, where each edge $e_{ij} = (v_i, v_j)$ connects two nodes, $v_i$ and $v_j$. The structure of a graph can be represented by its adjacency matrix $A \in \mathbb{R}^{n \times n}$, where $A_{ij} = 1$ if there is an edge between nodes $v_i$ and $v_j$, and 0 otherwise. This representation is limited to pairwise relationships between nodes. A hypergraph $\mathcal{G} = (\mathcal{V}, \mathcal{E})$, on the other hand, generalises the concept of a graph by allowing hyperedges $e_i \in \mathcal{E}$ to connect any subset of nodes, enabling the representation of complex, multi-node relationships. A hypergraph is represented by its incidence matrix $\mathcal{H} \in \mathbb{R}^{n \times m}$, where $n$ is the number of nodes,

$m$ is the number of hyperedges, and $\mathcal{H}_{ij} = 1$ if node $v_i$ is connected to hyperedge $e_j$, and 0 otherwise. This matrix effectively captures high-order interactions among nodes, making it suitable for modeling heterogeneous graphs. To convert a hypergraph into a homogeneous graph, we can compute a projected adjacency matrix $A \in \mathbb{R}^{n \times n}$ as $A = \mathcal{H}W\mathcal{H}^\top - D_v$, where $W \in \mathbb{R}^{m \times m}$ is the hyperedge weight matrix, and $D_v$ is the diagonal node degree matrix, with $D_v(i,i) = \sum_{j=1}^{m} \mathcal{H}_{ij}$. This transformation reduces the hypergraph to a traditional graph where two nodes are connected if they share common hyperedges.

**Graph Convolutional Networks (GCNs)** (Kipf & Welling, 2016) are designed to operate directly on graph-structured data, allowing for effective representation learning from the connections between nodes. In the context of a graph, the input consists of an adjacency matrix $A$ that encodes the relationships between nodes and a feature matrix $X$ that contains the features of each node. The fundamental operation in a GCN is performed through a series of graph convolutional layers. The update rule for node embeddings at layer $l$ is defined as $H^{(l+1)} = \sigma(\tilde{A}H^{(l)}W^{(l)})$, where $H^{(l)}$ represents the node embeddings at layer $l$ (for layer 0, $H^{(0)}$ is the input feature matrix), $W^{(l)}$ is the trainable weight matrix at layer $l$, $\sigma$ is a non-linear activation function (e.g., ReLU), and $\tilde{A}$ is the normalised adjacency matrix defined as $\tilde{A} = \hat{D}^{-1/2}\hat{A}\hat{D}^{-1/2}$. Here, $\hat{A} = A + I$ is the adjacency matrix with self-loops (adding the identity matrix $I$ to include each node's feature in the aggregation), and $\hat{D}$ is the degree matrix of $\hat{A}$, where $\hat{D}_{ii} = \sum_j \hat{A}_{ij}$. The term $\hat{D}^{-1/2}$ represents the normalised degree matrix (the square root inverse of the degree matrix). This normalisation ensures the node features are scaled properly, preventing exploding or vanishing gradients during training.

**Hypergraph Neural Network (HGNN)** is a powerful extension of the GCN designed to capture high-order relationships among nodes by leveraging the structure of hypergraphs. In a hypergraph $\mathcal{G}$, each hyperedge $e_j$ can connect multiple nodes from the set $\mathcal{V}$, allowing for more complex interactions than simple pairwise connections. HGNNs use the hypergraph Laplacian, derived from the node degree matrix $D_v$ and the hyperedge degree matrix $D_e$, to propagate information across the hypergraph. This is achieved through a message-passing mechanism that updates node features using the formula:

$$H^{(l+1)} = \sigma\left(D_v^{-1/2}\mathcal{H}WD_e^{-1}\mathcal{H}^T D_v^{-1/2}H^{(l)}\Theta^{(l)}\right) \tag{1}$$

where $\Theta^{(l)}$ represents the learnable weights and $\sigma(\cdot)$ is a non-linear activation function. This formulation allows HGNNs to effectively aggregate and propagate features, accounting for the structure of the hypergraph, thus enabling the modeling of complex dependencies between nodes. HGNNs have demonstrated their versatility and effectiveness in various applications, including recommendation systems, social networks, and biological data analysis (Feng et al., 2019).

## 3 METHODOLOGY

The overall architecture of the proposed DistillHGNN framework is shown in Figure 1. The proposed method aims to enhance the inference speed and memory efficiency of HGNNs while maintaining performance comparable to HGNNs. The method is structured around a teacher-student knowledge distillation Figure 1. The teacher model is formed by an HGNN and a MLP. The HGNN generates node embeddings $Z^t \in \mathbb{R}^{n \times d}$ by processing the node feature matrix $X \in \mathbb{R}^{n \times f}$ and the hypergraph incidence matrix $\mathcal{H} \in \mathbb{R}^{n \times m}$, where $n$ is the number of nodes and $m$ is the number of hyperedges. Given the node feature matrix $X$, the HGNN propagates these features through the hypergraph structure using the hypergraph Laplacian:

$$\mathcal{L} = D_v^{-1/2}\mathcal{H}WD_e^{-1}\mathcal{H}^T D_v^{-1/2} \tag{2}$$

Here, $D_v \in \mathbb{R}^{n \times n}$ is the diagonal node degree matrix, $D_e \in \mathbb{R}^{m \times m}$ is the diagonal hyperedge degree matrix, and $W \in \mathbb{R}^{m \times m}$ is the hyperedge weight matrix. The HGNN generates node embeddings $Z^t \in \mathbb{R}^{n \times d}$ by propagating the input features $X$ through the Laplacian as $Z^t = H^{(L)}$, where $L$ is the number of layers in the HGNN, the update rule is defined as:

$$H^{(l+1)} = \sigma\left(\mathcal{L}H^{(l)}\Theta^{(l)}\right) \tag{3}$$

The embedding $Z^t$ is then input into the teacher MLP to predict soft labels $Y^t$. Let $C = \{c_1, c_2, \ldots, c_k\}$ be a set of node classes. A soft label is a vector with a length equal to the number

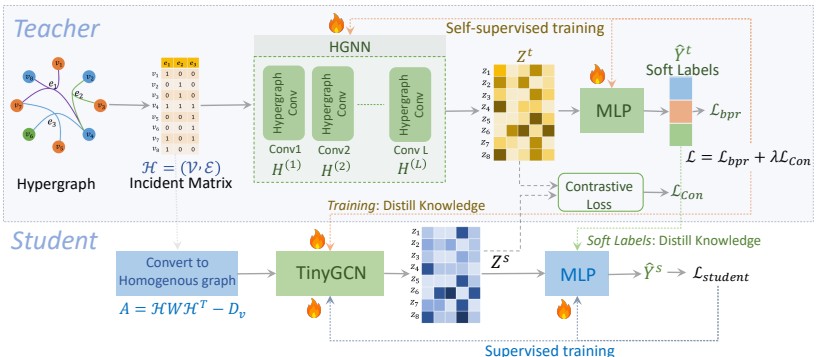

Figure 1: The proposed model consists of a teacher section (HGNN) and a student section (TinyGCN). The HGNN represents hypergraphs and generates soft labels using high-order relationships, while the TinyGCN captures direct neighborhoods relations in graphs. The soft labels guide the student model via a KL divergence term and are also used to generate labels for datasets with insufficient ground truth annotations. Both models are trained using a combination of supervised loss and contrastive loss to align their embeddings, enabling efficient knowledge distillation from teacher to student.

of classes (i.e., $|C|$), where each value in the $i$-th cell represents the probability of the node belonging to the class $c_i \in C$. The supervised loss, called Bayesian Personalised Ranking (BPR) loss, is defined as:

$$L_{\text{bpr}} = \frac{1}{|V^L|} \sum_{v \in V^L} \left( Y_v - Y_v^t \right)^2 \tag{4}$$

where $Y_v$ is the true label for labelled nodes $V^L$, and $Y_v^t$ is the soft label predicted for the node $v$. To transfer knowledge from the teacher to the student model, we propose a lightweight GCN called TinyGCN, designed to mimic the embedding generation capabilities of HGNN. TinyGCN simplifies the standard GCN architecture by removing non-linear activation functions and using only a single layer. This minimalistic design allows TinyGCN to effectively capture the high-order relationships learned by the teacher model while significantly reducing inference time and computational complexity. In parallel, the graph is also passed through the TinyGCN to generate embeddings $Z^s \in \mathbb{R}^{n \times d}$ using the node feature matrix $X$ and adjacency matrix $A^s \in \mathbb{R}^{n \times n}$ obtained from the incidence matrix $\mathcal{H}$. The node embeddings are updated using only the linear aggregation of neighbouring nodes' information via the adjacency matrix, represented as:

$$Z^s = \hat{A}^s X W^s \tag{5}$$

where $\hat{A}^s = A^s + I$ and $W^s$ is the trainable weight matrix. To ensure that the TinyGCN captures similar high-order relationships as the HGNN, a contrastive learning approach is applied using the InfoNCE loss:

$$L_{\text{con}} = -\frac{1}{|V|} \sum_{v \in V} \log \left( \frac{\exp \left( Z_v^s \cdot Z_v^t / \tau \right)}{\sum_{v' \in V} \exp \left( Z_{v'}^s \cdot Z_{v'}^t / \tau \right)} \right) \tag{6}$$

where $\tau$ is the temperature parameter that scales the similarity scores between embeddings. The total loss for the teacher model combines the supervised and contrastive losses as:

$$L_{\text{teacher}} = L_{\text{bpr}} + \gamma L_{\text{con}} \tag{7}$$

with $\gamma$ as a hyperparameter balancing the contributions of both losses. This ensures a seamless transfer of the topological structure and high-order information from the HGNN to the TinyGCN. Additionally, the teacher model generates soft labels for all nodes in the graph, representing the probability distribution over classes for each node. Given the limited availability of human-annotated labels, these soft labels serve as a valuable form of knowledge, transferred to the student model to provide richer supervisory signals. This approach helps the student model effectively learn the underlying data distribution, enabling it to perform well even with a sparse set of ground truth labels.

The student model includes the TinyGCN and a separate MLP. TinyGCN produces embeddings $Z^s$ for each node, which is then fed into the student MLP to predict the target labels. We use both labelled data and soft labels provided by the teacher model to train the student model. For a given node $v$, the predicted label is:

$$\hat{Y}_v^s = \text{MLP}^s(Z_v^s), \tag{8}$$

where $Z_v^s$ is obtained by applying TinyGCN using Equation (5). The loss function for the student model is:

$$L_{\text{student}} = \frac{1}{|V^L|} \sum_{v \in V^L} (\hat{Y}_v - Y_v)^2 + \lambda \frac{1}{|V|} \sum_{v \in V} \text{KL}(\hat{Y}_v || Y_v^t) \tag{9}$$

where $Y_v^t$ represents the soft label obtained from the teacher model, and KL denotes the Kullback-Leibler divergence, which measures the difference between the predicted probability distribution and the target distribution provided by the teacher. The parameter $\lambda$ controls the influence of the distillation loss in the overall training objective, balancing the impact of learning from the teacher's knowledge against other learning signals. This approach ensures that the student model effectively assimilates the nuanced class probabilities inferred by the teacher. The algorithm of the proposed method is provided in Algorithm 1 in the Appendix A.

## 4 EXPERIMENTS

### 4.1 DATASETS

In our experiments, we utilise eight well-known graph datasets. The Cora dataset, introduced by Sen et al. (Sen et al., 2008), and the Citeseer dataset, developed by Giles et al. (Giles et al., 1998), have been transformed into hypergraph datasets, namely CC-Cora and CC-Citeseer, by Yadati et al. (Yadati et al., 2019). In these two datasets, vertices represent academic papers, and hyperedges connect co-cited papers (CC). Each vertex is labelled according to the topic of its corresponding paper. We include the complete IMDB dataset and its subset IMDBAW from Fu et al. (Fu et al., 2019), which features multiple types of hyperedges: user-movie interactions, actor-movie collaborations, and director-movie relationships. While IMDBAW focuses on co-actor and co-director relationships, the complete IMDB encompasses a broader network of 142,129 nodes across movies, users, directors, and actors. Additionally, we incorporate the complete DBLP dataset and its three subsets: DBLP-Paper, DBLP-Term, and DBLP-Conf, as introduced by Sun et al. (Sun et al., 2011). The complete DBLP dataset contains 66,543 nodes spanning papers, authors, venues, and terms, with hyperedges formed through various academic relationships. In its subsets, hyperedges are formed based on collaborations (DBLP-Paper), the use of the same term (DBLP-Term), and papers published at the same conference (DBLP-Conf). The vertex labels correspond to the authors' research areas. A summary of the dataset statistics is provided in Table 3 in the Appendix B.

### 4.2 EVALUATION METHODS

We conducted each experiment five times using different random seeds, reporting both the average performance and standard deviation. Our evaluation is based on three training methods: transductive, inductive, and production (Tian et al., 2022; Feng et al., 2024). In the transductive method, the training and test data are used during evaluation, allowing the model to learn from the entire dataset. In contrast, the inductive method trains the model on one dataset and tests it on a completely different, unseen dataset. The production method combines both transductive and inductive predictions to simulate a more realistic deployment scenario. For each dataset, 20% of the data was used for validation, and 10% was used for testing. Accuracy was the primary metric for performance comparison, and the model with the best validation performance was applied to the test set for final evaluation.

### 4.3 BASELINE METHODS

We evaluate DistillHGNN against two different types of approaches. The first category includes graph-based methods, comprising traditional GNN models (Scarselli et al., 2008), GCN by Kipf and Welling (Kipf & Welling, 2016), MLP by Taud et al. (Taud & Mas, 2018), and knowledge distillation methods. These include GNN-to-MLP models like GLNN by Zhang et al. (Zhang et al., 2022)

and KRD by Wu et al. (Wu et al., 2023), which operate on graph structures. The second category comprises hypergraph-based methods. We utilise the Hypergraph Neural Network (HGNN) by Feng et al. (Feng et al., 2019) and the knowledge distillation method HGNN-to-MLP (LightHGNN) by Feng et al. (Feng et al., 2024). Accordingly, we generate two versions of the data: for the first type of methods, we use the graph structure, which includes GNN, MLP, and GLNN, while for the second type, we utilise the hypergraph structure for HGNN and LightHGNN. The baseline experiments and our methods are implemented using PyTorch and the Deep Hypergraph library [1]. We evaluate the proposed method for classification based on two key aspects: accuracy and inference speed. Table 1 presents the results based on accuracy across the three training methods.

Table 1: Experimental results on eight hypergraph datasets under production setting using transductive (Tran.), inductive (Ind.), and productive (Prod.) training methods.

| Dataset | Setting | GNN | GCN | MLP | HGNN | GLNN | KRD | LightHGNN | DistillHGNN |
|---|---|---|---|---|---|---|---|---|---|
| IMDB | Tran. | $46.45_{+2.15}$ | $46.92_{+1.65}$ | $43.22_{+1.95}$ | $51.45_{+1.77}$ | $45.88_{+2.33}$ | $47.33_{+1.88}$ | $50.22_{+1.95}$ | $\mathbf{51.88_{+1.66}}$ |
|  | Ind. | $47.33_{+2.44}$ | $47.67_{+1.72}$ | $44.15_{+2.12}$ | $52.28_{+1.95}$ | $46.85_{+2.55}$ | $48.22_{+2.05}$ | $51.18_{+2.12}$ | $\mathbf{52.33_{+1.77}}$ |
|  | Prod. | $46.88_{+2.33}$ | $47.15_{+1.88}$ | $43.77_{+2.05}$ | $51.22_{+1.88}$ | $46.12_{+2.44}$ | $47.88_{+1.95}$ | $50.45_{+2.05}$ | $\mathbf{51.92_{+1.66}}$ |
| IMDB-AW | Tran. | $46.26_{+1.36}$ | $46.77_{+2.12}$ | $42.70_{+1.67}$ | $\mathbf{53.36_{+2.25}}$ | $45.22_{+2.51}$ | $47.15_{+1.88}$ | $51.11_{+0.77}$ | $52.34_{+1.62}$ |
|  | Ind. | $48.28_{+4.43}$ | $48.67_{+2.33}$ | $43.35_{+3.33}$ | $54.12_{+2.74}$ | $46.50_{+2.23}$ | $49.12_{+2.55}$ | $52.27_{+1.14}$ | $\mathbf{55.27_{+1.16}}$ |
|  | Prod. | $47.33_{+1.82}$ | $47.88_{+1.95}$ | $43.15_{+1.92}$ | $53.31_{+3.01}$ | $45.16_{+3.98}$ | $48.22_{+2.15}$ | $51.84_{+3.51}$ | $\mathbf{53.93_{+2.14}}$ |
| CC-Citeseer | Tran. | $53.06_{+0.62}$ | $53.88_{+1.62}$ | $46.76_{+0.83}$ | $\mathbf{62.26_{+1.68}}$ | $52.57_{+1.73}$ | $54.12_{+1.55}$ | $59.65_{+2.12}$ | $61.02_{+1.62}$ |
|  | Ind. | $55.00_{+1.09}$ | $55.22_{+1.88}$ | $49.20_{+2.60}$ | $\mathbf{62.38_{+2.11}}$ | $54.45_{+3.01}$ | $55.67_{+2.12}$ | $61.36_{+1.27}$ | $62.18_{+2.16}$ |
|  | Prod. | $53.60_{+0.34}$ | $54.05_{+2.01}$ | $47.20_{+1.60}$ | $61.39_{+3.11}$ | $52.08_{+2.55}$ | $54.33_{+1.92}$ | $60.11_{+1.63}$ | $\mathbf{61.88_{+2.14}}$ |
| CC-Cora | Tran. | $52.26_{+3.25}$ | $52.88_{+1.95}$ | $45.16_{+2.51}$ | $\mathbf{65.17_{+1.68}}$ | $51.22_{+1.10}$ | $53.45_{+2.15}$ | $63.65_{+2.12}$ | $65.02_{+1.62}$ |
|  | Ind. | $54.39_{+2.15}$ | $54.67_{+2.05}$ | $48.66_{+2.19}$ | $\mathbf{66.88_{+2.11}}$ | $53.50_{+1.81}$ | $55.12_{+1.88}$ | $65.76_{+1.27}$ | $66.78_{+1.16}$ |
|  | Prod. | $54.15_{+3.44}$ | $54.55_{+1.92}$ | $48.02_{+2.05}$ | $65.52_{+2.11}$ | $53.19_{+2.75}$ | $54.88_{+2.33}$ | $64.11_{+1.63}$ | $\mathbf{65.68_{+2.14}}$ |

| Dataset | Setting | GNN | GCN | MLP | HGNN | GLNN | KRD | LightHGNN | DistillHGNN |
|---|---|---|---|---|---|---|---|---|---|
| DBLP | Tran. | $75.33_{+2.15}$ | $75.92_{+1.88}$ | $66.88_{+2.77}$ | $\mathbf{83.26_{+1.55}}$ | $72.45_{+2.55}$ | $76.45_{+1.95}$ | $81.45_{+2.88}$ | $82.88_{+1.66}$ |
|  | Ind. | $76.15_{+2.44}$ | $76.67_{+1.95}$ | $67.22_{+3.15}$ | $84.12_{+1.77}$ | $73.12_{+2.88}$ | $77.22_{+2.12}$ | $82.33_{+2.55}$ | $\mathbf{84.45_{+1.88}}$ |
|  | Prod. | $75.88_{+2.33}$ | $76.15_{+1.92}$ | $66.45_{+2.95}$ | $83.55_{+1.88}$ | $72.88_{+2.66}$ | $76.88_{+2.05}$ | $81.88_{+2.44}$ | $\mathbf{83.77_{+1.75}}$ |
| DBLP-Paper | Prod. | $65.15_{+2.21}$ | $65.88_{+1.95}$ | $60.47_{+2.44}$ | $\mathbf{71.80_{+1.08}}$ | $62.42_{+4.02}$ | $66.33_{+2.15}$ | $69.12_{+3.22}$ | $71.22_{+2.55}$ |
|  | Tran. | $66.23_{+1.55}$ | $66.77_{+2.12}$ | $62.38_{+2.45}$ | $72.52_{+0.98}$ | $63.87_{+3.76}$ | $67.12_{+2.33}$ | $71.22_{+2.63}$ | $\mathbf{72.02_{+1.76}}$ |
|  | Ind. | $65.57_{+1.86}$ | $66.22_{+2.05}$ | $61.50_{+3.07}$ | $72.08_{+1.54}$ | $63.17_{+3.22}$ | $66.88_{+1.92}$ | $70.69_{+2.17}$ | $\mathbf{71.16_{+1.64}}$ |
| DBLP-Term | Prod. | $66.33_{+2.31}$ | $66.92_{+1.88}$ | $61.20_{+3.65}$ | $\mathbf{72.10_{+0.90}}$ | $64.15_{+2.84}$ | $67.45_{+2.15}$ | $70.16_{+3.29}$ | $71.67_{+0.87}$ |
|  | Tran. | $67.52_{+3.29}$ | $67.88_{+2.15}$ | $62.12_{+3.50}$ | $73.46_{+1.63}$ | $65.24_{+2.72}$ | $68.33_{+2.44}$ | $71.64_{+1.53}$ | $\mathbf{73.79_{+1.32}}$ |
|  | Ind. | $66.13_{+4.06}$ | $66.77_{+2.05}$ | $61.68_{+2.88}$ | $\mathbf{73.12_{+1.33}}$ | $64.87_{+3.15}$ | $67.22_{+2.33}$ | $71.51_{+2.17}$ | $72.45_{+1.76}$ |
| DBLP-Conf | Prod. | $72.30_{+2.55}$ | $72.88_{+1.95}$ | $63.53_{+2.77}$ | $\mathbf{90.03_{+1.66}}$ | $69.18_{+3.11}$ | $73.45_{+2.15}$ | $88.10_{+3.51}$ | $89.27_{+2.17}$ |
|  | Tran. | $74.18_{+2.84}$ | $74.67_{+2.12}$ | $65.12_{+3.44}$ | $\mathbf{92.22_{+2.78}}$ | $71.45_{+2.23}$ | $75.22_{+2.33}$ | $90.74_{+4.20}$ | $92.11_{+1.84}$ |
|  | Ind. | $74.26_{+2.62}$ | $74.88_{+2.05}$ | $64.22_{+4.07}$ | $91.00_{+2.42}$ | $71.02_{+2.96}$ | $75.33_{+1.92}$ | $90.05_{+4.04}$ | $\mathbf{91.38_{+3.25}}$ |

The DistillHGNN model demonstrates substantial advantages in classification tasks by utilising knowledge distillation to enhance its performance over traditional graph-based and hypergraph-based methods. As shown in Table 1, DistillHGNN consistently achieves either the highest or second-highest accuracy across various datasets and settings, including training (Tran), inductive (Ind), and production (Prod) scenarios. Although DistillHGNN's accuracy is slightly lower than HGNN's top performance, it consistently outperforms LightHGNN and shows a small but noticeable improvement in accuracy. Additionally, DistillHGNN significantly surpasses traditional models such as GNN, GCN, MLP, GLNN and KRD.

## 4.4 Balancing accuracy and inference time

In the experiments, we evaluate DistillHGNN against other models to analyse the trade-off between predictive performance and computational efficiency. For all subsequent experiments, we employ the production method for training. Additionally, we use three versions of the DBLP dataset (based on Papers, Terms, and Conferences), running the model on all three variants and taking the average to ensure consistency. This averaged result is referred to as the DBLP dataset in our analysis.

---

[1] https://github.com/iMoonLab/DeepHypergraph

In this evaluation phase, we focus on two datasets—IMDB-AW and DBLP—and compare Distill-HGNN with other models by examining the trade-off between model accuracy and inference time. This approach allows us to evaluate both the effectiveness and computational efficiency of each model, offering a comprehensive comparison across these dimensions. In real-world applications, an ideal model should maintain high accuracy while minimising inference time, particularly when computational resources and processing speed are critical. Models with high accuracy but slow inference times, such as GNNs or HGNNs, may not be practical for time-sensitive applications, while faster models like MLPs, which sacrifice accuracy, may fail to meet performance expectations. The strength of knowledge distillation models lies in achieving a balanced trade-off, offering both competitive accuracy and reasonable computational efficiency. Higher accuracy often comes at the expense of longer inference time, but DistillHGNN is designed to balance these factors. The results, illustrated in Figure 2, demonstrate how each model's accuracy compares with its inference time across the datasets.

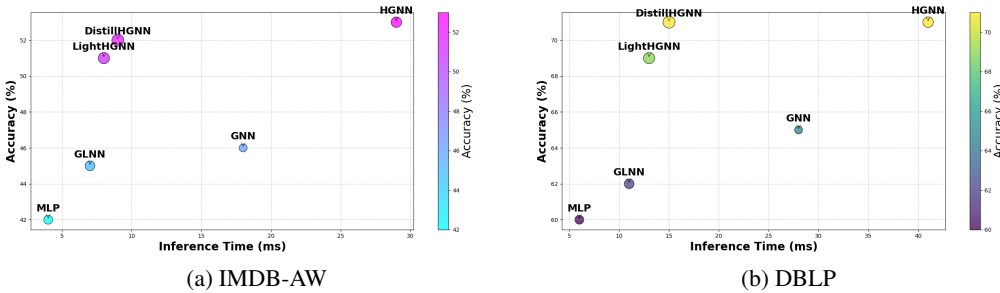

(a) IMDB-AW                                                (b) DBLP

Figure 2: Comparison of model accuracy and inference time across different methods on the IMDB-AW and DBLP datasets. The figure illustrates the trade-off between predictive performance and computational efficiency, highlighting the balance achieved by DistillHGNN.

Figure 2 demonstrates that DistillHGNN significantly reduces inference time compared to HGNN, achieving a reduction of approximately 69%. This improvement underscores DistillHGNN's efficiency, as it maintains high accuracy while providing much faster inference. Although MLP offers the quickest inference time at just 4 ms, its accuracy of 0.42 makes it less suitable for high-performance applications. DistillHGNN's inference time is not only significantly faster than HGNN's but also comparable to LightHGNN's, while delivering superior accuracy. This indicates that the knowledge distillation process enhances overall performance without adding substantial computational overhead. Additionally, DistillHGNN outperforms GNN and GLNN in accuracy, making it an effective model for large-scale tasks that require a balance of performance and speed. Overall, DistillHGNN effectively balances high accuracy with computational efficiency, making it an ideal choice for scenarios where both model performance and rapid execution are crucial, such as in real-time systems or large-scale applications.

## 4.5 ACCURACY OF THE METHODS

A key challenge in knowledge distillation is to ensure that the student captures not only direct relationships between nodes but also crucial high-order relations. In this process, we focus on two relationship types within the student model: direct (neighbouring) relations and high-order relations. The teacher model, leveraging a hypergraph structure, excels at capturing high-order relationships but often suffers from high inference times due to the complexity of hyperedges. Previous research (Tian et al., 2022; Wu et al., 2023; Feng et al., 2024), has utilised soft labels to facilitate knowledge transfer. However, while soft labels approximate the teacher's behaviour, they alone cannot fully convey the high-order structural knowledge captured by the hypergraph. To overcome this limitation, we propose a combined approach that utilises both soft labels and structural knowledge from HGNN as the teacher model. To implement this combined distillation, we employ contrastive learning within the TinyGCN framework. The student model receives two types of inputs: one based on the simpler TinyGCN architecture and the other derived from high-order embeddings generated by the HGNN teacher model. To evaluate the effectiveness of DistillHGNN, we propose the High-Order Preservation Score ($H_{\text{pres}}$), which quantifies how well the student model retains the complex, high-order relationships transferred from the teacher model. DistillHGNN addresses this challenge

by incorporating both soft labels and high-order relation embeddings through contrastive learning. $H_{\text{pres}}$ is computed as follows:

$$H_{\text{pres}} = \frac{1}{|V|} \sum_{v \in V} \frac{1}{|N(V)|} \sum_{u \in N(v)} \text{Sim}(\mathbf{x}_u^{\text{teacher}}, \mathbf{x}_u^{\text{student}}) \tag{10}$$

Here, $|V|$ is the total number of nodes, $N(v)$ is the set of $k$-nearest neighbours of node $v$, and $\mathbf{x}_u^{\text{teacher}}$ and $\mathbf{x}_u^{\text{student}}$ are the embeddings generated by the teacher and student models, respectively. The similarity measure Sim can be based on cosine similarity or Euclidean distance, depending on the experimental setup. This score evaluates the extent to which the student model replicates the high-order structural information encoded in the teacher model, focusing on the similarity between the embeddings of neighbouring nodes in both models. The results indicate in Table 2 as follows: The results demonstrate a clear progression in the High-Order Relation Preservation Score ($H_{\text{pres}}$)

Table 2: Comparison of knowledge distillation methods based on the preservation score.

| Method | CC-Cora | CC-Citeseer | IMDB-AW | DBLP | *Mean* |
|---|---|---|---|---|---|
| GLNN (Tian et al., 2022) | 0.54 | 0.58 | 0.51 | 0.67 | **0.575** |
| KRD (Wu et al., 2023) | 0.56 | 0.62 | 0.54 | 0.71 | **0.6075** |
| LightHGNN (Feng et al., 2024) | 0.72 | 0.78 | 0.74 | 0.83 | **0.7675** |
| DistillHGNN | 0.78 | 0.84 | 0.81 | 0.88 | **0.8275** |

across the evaluated methods, with DistillHGNN achieving the highest mean score of 0.8275. This indicates that methods integrating greater structural knowledge are significantly more effective at preserving high-order relationships.

## 4.6 INFERENCE TIME AND MEMORY EFFICIENCY ANALYSIS

Distilling knowledge to MLPs offers faster execution times and lower memory usage but typically result in reduced accuracy. In contrast, HGNNs deliver superior performance, though at the cost of slower inference times and higher memory consumption. DistillHGNN successfully bridges this gap by achieving a balanced trade-off between memory usage, storage, and accuracy. As illustrated in Figure 2, DistillHGNN maintains competitive inference times and reduces memory consumption through its lightweight student model design without sacrificing accuracy. The model also optimises memory storage, ensuring that it utilises less disk space compared to traditional HGNNs by pruning less critical connections while retaining important structural information. By utilising heterogeneous graphs, such as those derived from the IMDB and DBLP datasets (Fan et al., 2021), the model benefits from the rich, multi-relational structures present in the data. These datasets include complex relationships where nodes—such as movies or academic papers—interact with various entities (e.g., users, directors, authors), forming hyperedges that capture higher-order connections. Detailed statistics of the datasets and the results from this evaluation are presented in Tables 4 and 5, respectively, in Appendix C.

## 4.7 ABLATION STUDY

In this section, we present findings that evaluate the effectiveness of different knowledge transfer methods employed in the DistillHGNN framework. The proposed model is assessed using three distinct approaches: (1) soft labels alone, (2) structural knowledge alone, and (3) a combination of both (DistillHGNN). The results are illustrated in Figure 3, which includes calculations of both accuracy and inference time for each method. The accuracy scores reveal a significant improvement when using the combined approach of DistillHGNN, which leverages both soft labels and structural knowledge, compared to using either method individually. Across all datasets, the combined method consistently outperforms the individual approaches, with accuracy improvements ranging from 3% to 10%. These findings underscore the advantages of utilising both soft labels and structural knowledge within the DistillHGNN framework, demonstrating a favourable trade-off between speed and accuracy.

In DistillHGNN, knowledge transfer occurs through both structural knowledge and soft labels from the teacher model to the student model. When CL is incorporated, the model achieves a more refined

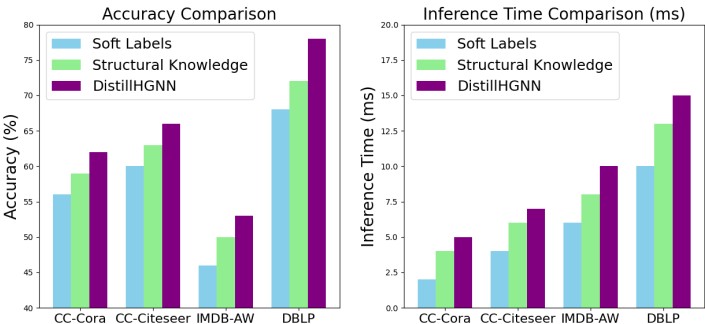

Figure 3: The proposed model is evaluated using three different knowledge transfer methods: (1) soft labels alone, (2) structural knowledge alone, and (3) a combination of both (DistillHGNN). The evaluation includes calculating both accuracy and inference time for each method.

alignment of node embeddings, enabling the student to effectively capture intricate high-order relationships. The results comparing models with and without CL are presented in Figure 4 as follows: The accuracy comparison between the models with and without contrastive learning across datasets

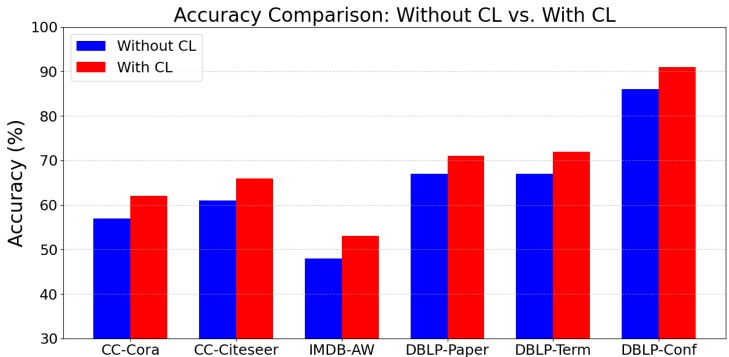

Figure 4: The proposed model is evaluated based on the absence of contrastive learning (lack of CL) and with contrastive learning (DistillHGNN), focusing on accuracy across four datasets.

clearly shows performance improvements when using CL. These results indicate that incorporating both structural knowledge and contrastive learning in DistillHGNN enhances generalisation and performance, particularly in more complex datasets like DBLP. Despite slightly slower inference times, the accuracy gains justify the use of DistillHGNN for achieving more accurate predictions.

To further evaluate the performance of DistillHGNN, we examine the impact of increasing the number of layers in the proposed model on both accuracy and inference time. This analysis helps determine the influence of model depth on the quality of predictions and computational efficiency components of the proposed model. The results of this evaluation are presented in Appendix D. The proposed method includes several hyperparameters that play key roles in controlling trade-offs between different components, the learning process, and overall performance. Table 6 shows the results of the evaluation of the DistillHGNN framework based on different configurations of hyperparameters and their corresponding accuracy metrics in Appendix E. In this experiment, we evaluate the performance of DistillHGNN across six hypergraph datasets under different training sizes. The goal is to observe how varying the proportion of training data affects the model's accuracy. We compare the DistillHGNN model with several baselines: GNNs, MLP, GLNN, HGNN, and LightHGNN. The results are presented in Table 7 in Appendix F. To thoroughly assess the performance of DistillHGNN, we conducted an ablation study focusing on the impact of different training durations, specifically varying the number of training epochs. This evaluation aims to determine how the duration of training influences the model's accuracy and generalisation capability across different datasets. Each model was trained using the same dataset and hyperparameter settings to ensure a fair

comparison. The results of this study are shown in Figure 5, which presents the accuracy achieved by DistillHGNN for each training epoch setting.

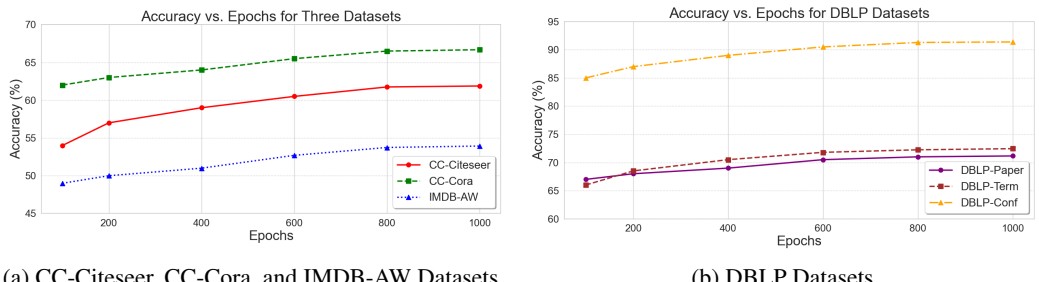

(a) CC-Citeseer, CC-Cora, and IMDB-AW Datasets          (b) DBLP Datasets

Figure 5: Evaluation of the proposed method's performance across different numbers of epochs, ranging from 100 to 1000. This analysis examines the impact of varying training epochs on model performance, offering insights into the optimal training duration for achieving the best accuracy.

The experimental results indicate that as the number of epochs increases, model accuracy generally improves, though at varying rates depending on the dataset. This suggests that the datasets contain rich features that the model can effectively leverage, benefiting significantly from additional training epochs. Next, we conducted a comprehensive visual analysis of DistillHGNN's performance using multiple visualisation techniques to evaluate the effectiveness of knowledge transfer between the teacher and student models. The results of this evaluation are presented in Appendix G.

## 5  CONCLUSION

In this paper, we address two key challenges in knowledge distillation for graph-based models: (1) effectively transferring high-order relationships between nodes, and (2) overcoming the limitations of using soft labels as the sole medium for transferring knowledge from the teacher model to the student model. To address the first challenge, we leveraged a hypergraph structure within the teacher model, allowing for the capture and transfer of high-order relationships that go beyond direct node connections. For the second challenge, we employed the contrastive learning (CL) framework in combination with soft labels to enhance the knowledge transfer process. The contrastive learning mechanism ensures that the embeddings generated by the student model align with those of the teacher model, thereby preserving the high-order relational knowledge encoded in the HGNN. The experimental results demonstrate that the proposed approach achieves superior accuracy compared to traditional knowledge distillation methods. Specifically, the use of hypergraph-based embeddings and the integration of soft label distillation with structural knowledge led to better performance in terms of both classification accuracy and the preservation of high-order node relations. Moreover, our method strikes a desirable balance between inference speed and memory efficiency. TinyGCN significantly reduces computational complexity, its performance remains comparable to that of the more complex HGNN model. Overall, this approach opens up new possibilities for lightweight models that retain the expressiveness of more complex networks while ensuring faster inference, making it suitable for real-world applications where computational resources are limited.

### ACKNOWLEDGEMENTS

This research is partially supported by Australian Research Council through projects DP240100963, LP230100439, and IM240100042.

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

## A  ALGORITHM

The proposed DistillHGNN framework is a knowledge distillation approach that leverages the power of a HGNN as the teacher model and a lightweight TinyGCN as the student model. The goal of this framework is to transfer knowledge from the HGNN, which captures high-order relations among nodes through hyperedges, to the TinyGCN, which is designed for direct and low-complexity node aggregation. By distilling the rich information from the teacher model into the simpler student model, the system ensures efficient learning while maintaining competitive performance in tasks such as node classification. The distillation process includes the generation of soft labels from the teacher model, which are used as targets for the student model in conjunction with the true labels. Additionally, a contrastive loss function ensures that the embeddings produced by the student model align with the high-order relational embeddings learned by the teacher, reinforcing the transfer of meaningful information. The combination of BPR loss, contrastive loss, and Kullback-Leibler (KL) divergence enables a robust learning process for the student model. The overall procedure of the proposed DistillHGNN method is outlined in Algorithm 1.

## B  DATASETS

In this work, we evaluate the performance of the proposed DistillHGNN framework on several widely used benchmark datasets. These datasets come from various domains, including citation networks, movie databases, and bibliographic datasets, providing a diverse range of graph structures and node features. The IMDB dataset represents a comprehensive heterogeneous network from the Internet Movie Database, containing 142,129 nodes across four different types: movies (40,635 nodes), users (2,113 nodes), directors (4,060 nodes), and actors (95,321 nodes). The heterogeneous nature is reflected in its three types of relationships: user-movie interactions (1,216,358 edges), director-movie connections (15,732 edges), and actor-movie collaborations (364,058 edges). With a high average degree of 22.46, this dataset exhibits very dense connectivity patterns, making it particularly challenging for graph learning tasks.

The DBLP dataset is a heterogeneous bibliographic network comprising 66,543 nodes of four types: papers (43,128 nodes), authors (14,475 nodes), venues (20 nodes), and terms (8,920 nodes). The heterogeneity is manifested in its diverse edge types: author-paper collaborations (58,592 edges), venue-paper publications (20,770 edges), and term-paper associations (195,462 edges). With an

---

**Algorithm 1** DistillHGNN: HGNN with Knowledge Distillation

---

**Input:** Hypergraph $\mathcal{G} = \{\mathcal{V}, \mathcal{E}\}$, features $X$, incidence matrix $\mathcal{H}$, labelled data $D_L = \{V_L, Y_L\}$, number of epochs $epochs$, parameters $\tau, \gamma, \lambda$
**Output:** Student model parameters

1: **Initialise:** Model parameters for HGNN ($\Theta^{(l)}$), $MLP^t$ ($\Theta^t$), TinyGCN ($W^s$), $MLP^s$ ($\Theta^s$), and $H^{(0)}$
    **Step 1: Compute Laplacian and Adjusted Adjacency**
2: $\mathcal{L} \leftarrow D_v^{-1/2} \mathcal{H} W D_e^{-1} \mathcal{H}^\top D_v^{-1/2}$
3: $\hat{A}^s \leftarrow A^s + I$
    **Step 2: Pre-train the Teacher Model**
4: **for** epoch $= 1$ to $epochs$ **do**
5:     Compute HGNN embeddings: $Z^t \leftarrow H^{(L)}$, where

$$H^{(l+1)} = \sigma(\mathcal{L} H^{(l)} \Theta^{(l)})$$

6:     Generate teacher predictions: $Y^t \leftarrow \text{MLP}^t(Z^t)$
7:     Compute TinyGCN embeddings: $Z^s \leftarrow \hat{A}^s X W^s$
8:     Compute teacher loss:

$$L_{\text{teacher}} \leftarrow \frac{1}{|V_L|} \sum_{v \in V_L} (Y_v - Y_v^t)^2 - \gamma \frac{1}{|V|} \sum_{v \in V} \log \frac{\exp(Z_v^s \cdot Z_v^t / \tau)}{\sum_{v'} \exp(Z_{v'}^s \cdot Z_{v'}^t / \tau)}$$

9:     Update teacher parameters $\Theta^t$
10: **end for**
    **Step 3: Train the Student Model**
11: **for** epoch $= 1$ to $E$ **do**
12:     Generate soft labels ($Y^t$) for all nodes using frosen teacher.
13:     Compute student outputs:

$$Z^s \leftarrow \hat{A}^s X W^s, \quad \hat{Y}_v^s \leftarrow \text{MLP}^s(Z^s)$$

14:     Compute student loss:

$$L_{\text{student}} \leftarrow \frac{1}{|V_L|} \sum_{v \in V_L} (\hat{Y}_v^s - Y_v)^2 + \lambda \frac{1}{|V|} \sum_{v \in V} \text{KL}(\hat{Y}_v^s \| Y_v^t)$$

15:     Update student parameters $\{W^s, \Theta^s\}$
16: **end for**
17: **Return:** Student model parameters

---

average degree of 8.26, it presents a dense, interconnected structure while maintaining clear hierarchical relationships among different node types. CC-Citeseer and CC-Cora are standard citation network datasets where nodes represent research papers, and edges represent citation links between papers. These datasets are characterised as homogeneous due to their uniform node and edge types. Each paper (node) is represented by a bag-of-words feature vector, and the goal is to classify papers into different research topics. Their relatively low average degrees (3.2 and 3.8 respectively) indicate sparse connectivity patterns. IMDB-AW is a subset of the complete IMDB dataset that focuses on actor collaborations and relationships within the movie industry. Nodes represent actors, and edges connect actors who have appeared in the same movie. Despite being smaller than the complete IMDB dataset, it maintains its heterogeneous characteristics with an average degree of 8.4, indicating dense connectivity patterns.

DBLP-paper, DBLP-term, and DBLP-Conf are subsets of the complete DBLP bibliographic dataset, each highlighting different aspects of the academic network. These subsets maintain the heterogeneous nature of the complete dataset but with varying connectivity patterns:

1. DBLP-paper exhibits moderate connectivity (degree 5.2) focusing on paper-centric relationships.

2. DBLP-term shows higher connectivity (degree 7.1) emphasising term-paper associations.

3. DBLP-Conf demonstrates sparse but hierarchical structure (degree 284.2) concentrating on conference-paper relationships.

The summary of the dataset statistics, including the number of nodes, edges (or hyperedges), features, and classes, is provided in Table 3. This diverse collection of datasets, ranging from sparse homogeneous to very dense heterogeneous networks, allows for comprehensive evaluation of hypergraph-based methods across different network structures and application domains.

Table 3: Datasets used in the experiments.

| Dataset | Statistics | | | | Characteristics |
|---|---|---|---|---|---|
| | #Nodes | #Edges | #Feat | #Class | |
| DBLP | 66,543 | 274,824 | 334 | 4 | Dense, heterogeneous (deg=8.26) |
| IMDB | 142,129 | 1,596,148 | 3,066 | 3 | Very dense, heterogeneous (deg=22.46) |
| CC-Citeseer | 3,312 | 1,004 | 3,703 | 6 | Sparse, homogeneous (deg=3.2) |
| CC-Cora | 2,708 | 1,483 | 1,433 | 7 | Mod. sparse, homogeneous (deg=3.8) |
| IMDB-AW | 5,355 | 6,811 | 3,066 | 3 | Dense, heterogeneous (deg=8.4) |
| DBLP-paper | 14,376 | 14,475 | 334 | 4 | Moderate, heterogeneous (deg=5.2) |
| DBLP-term | 14,376 | 13,789 | 334 | 4 | High connect., heterogeneous (deg=7.1) |
| DBLP-Conf | 14,376 | 1,612 | 334 | 4 | Sparse, hierarchical (deg=284.2) |

## C COMPARISON OF DISTILLHGNN AND HGNN BASED ON INFERENCE TIME AND MEMORY USAGE

In this evaluation phase, we assess both the inference time and memory efficiency of the Distill-HGNN model using two heterogeneous datasets: IMDB and DBLP. These datasets are particularly challenging due to their diverse node types and complex relationships, making them ideal for evaluating both the effectiveness and computational performance of our model. The IMDB dataset comprises user, movie, director, and actor nodes, interconnected through various hyperedges, including user-movie, director-movie, and actor-movie relationships. This structure exemplifies a typical heterogeneous network where multiple node types and interactions are represented. Similarly, the DBLP dataset includes paper, author, venue, and term nodes, connected through their respective relationships (author-paper, venue-paper, and term-paper), representing another common structure found in academic collaboration networks. Table 4 summarises the key statistics of these datasets, including the number of nodes, types of nodes, and hyperedges. The complexity and scale of these datasets pose significant challenges in terms of inference time, making them ideal for evaluating DistillHGNN's ability to balance accuracy and computational efficiency.

Table 4: Statistics of the Datasets

| Dataset | Nodes | Count of Nodes | Hyperedges | Count of Edges |
|---|---|---|---|---|
| **IMDB** | Movie | 40,635 | | |
| | User | 2,113 | User-Movie | 1,216,358 |
| | Director | 4,060 | Director-Movie | 15,732 |
| | Actor | 95,321 | Actor-Movie | 364,058 |
| | **Total** | 142,129 | | 1,596,148 |
| **DBLP** | Paper | 43,128 | | |
| | Author | 14,475 | Author-Paper | 58,592 |
| | Venue | 20 | Venue-Paper | 20,770 |
| | Term | 8,920 | Term-Paper | 195,462 |
| | **Total** | 66,543 | | 274,824 |

Given the intricacies of these datasets, it is crucial to evaluate the inference time of DistillHGNN in comparison to the original HGNN model. This section aims to assess the inference performance of DistillHGNN on both IMDB and DBLP graphs, contrasting it with the performance of the standard HGNN model. The results, which include a comparative analysis of the inference times, are summarised in Table 5. These findings illustrate the efficiency of DistillHGNN in reducing inference time while maintaining competitive performance.

Table 5: Comparison of Inference Time (ms) between DistillHGNN and HGNN for IMDB and DBLP Datasets at Different Node Levels.

| Dataset | Hyperedge Count | HGNN | DistillHGNN | Improvement (%) |
|---------|-----------------|------|-------------|-----------------|
| **IMDB** | 10,000 | 8.12 | 0.67 | 12.12 |
| | 20,000 | 37.35 | 1.13 | 33.05 |
| | 30,000 | 84.70 | 1.68 | 50.42 |
| | 40,635 | 175.56 | 2.23 | 78.70 |
| **DBLP** | 10,000 | 9.47 | 0.72 | 13.15 |
| | 20,000 | 42.15 | 0.98 | 43.01 |
| | 30,000 | 75.33 | 1.51 | 49.88 |
| | 43,128 | 168.84 | 2.06 | 81.97 |

These results highlight the scalability and computational efficiency of DistillHGNN, especially when managing large-scale graphs. For example, with 40,635 nodes in the IMDB dataset, DistillHGNN demonstrates an impressive speed improvement, being almost 79 times faster than the HGNN model. Similarly, for the DBLP dataset with 43,128 nodes, DistillHGNN is nearly 82 times faster. The substantial speedup, particularly at larger node levels, underscores the practical advantages of Distill-HGNN in applications that require rapid inference on complex, multi-relational data. Furthermore, the consistently higher times-faster ratios across both datasets accentuate DistillHGNN's effectiveness in minimising computational overhead compared to the HGNN model. Overall, the analysis illustrates that DistillHGNN not only maintains competitive performance but also significantly enhances inference speed, making it a compelling choice for real-world applications that demand efficiency without sacrificing accuracy. This capability is particularly valuable in scenarios where the processing of vast amounts of data is required, such as recommendation systems and social network analyses.

## C.1 MEMORY EFFICIENCY ANALYSIS

This subsection provides a comprehensive analysis of the memory efficiency of DistillHGNN in comparison to traditional GNNs and MLP-based models (e.g., GLNN, KRD, and LightHGNN). While we do not claim that DistillHGNN achieves better inference time than other distillation models such as LightHGNN or GLNN, we demonstrate that it offers a favourable trade-off between memory efficiency and model performance. Here, we compare the memory required for all models during inference. GLNN, KRD, LightHGNN, and DistillHGNN are knowledge distillation methods. GLNN and KRD use a GNN as their teacher model and an MLP with $L$ layers as the student model. Both LightHGNN and DistillHGNN use an HGNN as their teacher. However, while LightHGNN distills knowledge into an MLP-based student model with $L$ layers, DistillHGNN transfers knowledge to a single-layer GNN (TinyGCN) and an MLP with $L$ layers. Since our focus is on comparing the memory efficiency of models during inference, we do not account for the memory required for the teacher model, which is trained offline.

Let's consider a uniform feature dimension $d$ across all models and node embeddings. A typical message-passing GNN updates node representations using the equation $H^{(l+1)} = \sigma(AH^{(l)}W^{(l)})$, where $H^{(l)} \in \mathbb{R}^{N \times d}$ is the node feature matrix at layer $l$, $A \in \mathbb{R}^{N \times N}$ is the adjacency matrix (which can be stored in a sparse format), and $W^{(l)} \in \mathbb{R}^{d \times d}$ is the trainable weight matrix. The memory complexity of a GNN primarily arises from storing node embeddings, adjacency information, model parameters, and intermediate activations. The node embeddings require $O(Nd)$ memory at each layer, while the adjacency matrix, if stored in a dense format, requires $O(N^2)$. However, for large and sparse graphs, storing only the nonzero elements results in $O(|E|)$ memory usage. The weight

matrices require $O(Ld^2)$ memory across $L$ layers, which remains relatively small compared to the node and adjacency storage. Additionally, storing activations from each layer during inference adds an extra $O(LNd)$ memory requirement. Thus, the total memory complexity of a GNN with $L$ layers is is $O(L(|E| + Nd + d^2))$.

Since all knowledge distillation models use MLPs as their student models, we compute the memory required for the MLP during inference. An MLP is a fully connected neural network where each layer performs a linear transformation followed by a non-linear activation function. The forward propagation of an MLP with $L$ layers is given by $H^{(l+1)} = \sigma(H^{(l)}W^{(l)} + b^{(l)})$, where $H^{(l)} \in \mathbb{R}^{N \times d}$ is the activation matrix at layer $l$, $W^{(l)} \in \mathbb{R}^{d \times d}$ is the trainable weight matrix, and $b^{(l)} \in \mathbb{R}^d$ is the bias vector. The memory complexity of an MLP consists of storing model parameters (weights and biases) and activations. The weight matrices across $L$ layers require $O(Ld^2)$ memory, while storing activations during inference requires $O(Ld)$. Note that during inference, memory required for backpropagation is not used. Thus, the total memory complexity of an MLP with $L$ layers to process the embedding of a node is $O(Ld + Ld^2)$. Therefore, the memory required during inference for GLNN, KRD, and LightHGNN, as they use only an MLP in their student model, is $O(Ld + Ld^2)$. Moreover, since our model incorporates a single-layer GCN (TinyGCN), it requires additional memory of $O(|E| + Nd + d^2)$.

## D EVALUATION OF DISTILLHGNN BASED ON THE NUMBER OF LAYERS

To further assess the performance of DistillHGNN, we evaluate the impact of increasing the number of layers in various sections of the proposed model. This includes the HGNN and Teacher MLP in the Teacher section, as well as TinyGCN and Distill MLP in the Student section, with a focus on accuracy and inference time. The selected layers for the proposed method include three layers for both HGNN and Teacher MLP, a single layer for TinyGCN, and two layers for Distill MLP. In this section, we experiment with each model by testing different configurations with multiple layers (specifically, 2, 3, 4, and 5 layers) and comparing the results based on accuracy and inference time. Our experiments are conducted on the IMDB-AW dataset, and the results are illustrated in Figure 6 as follows:

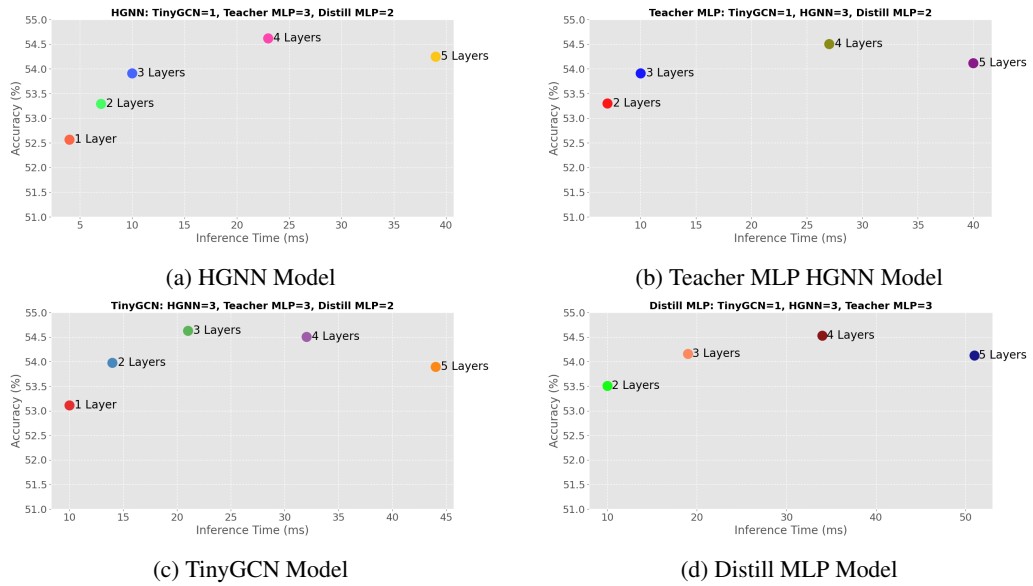

Figure 6: Evaluation of the Proposed Method Based on the Depth of Layers Configuration for HGNN, Teacher MLP, TinyGCN, and Distilled MLP using the IMDB-AW Dataset

In evaluating the performance of DistillHGNN, as indicated in Figure 6, we analysed the impact of increasing the number of layers across different sections of the proposed model, including the HGNN and Teacher MLP in the Teacher section, as well as TinyGCN and Distill MLP in the Student

section. The analysis focused on accuracy and inference time, based on the results obtained. The selected layers for the proposed method include three layers for both HGNN and Teacher MLP, a single layer for TinyGCN, and two layers for Distill MLP. This selection is driven by the need to balance model complexity with computational efficiency while ensuring optimal performance. The HGNN model demonstrates a consistent upward trend in accuracy as the number of layers increases, starting from 52.37% with 1 layer and reaching 54.15% with 5 layers. However, the decision to limit the depth of the HGNN to three layers is based on observed diminishing returns in accuracy beyond this point, as indicated by the marginal improvement from 54.12% at 4 layers to 54.15% at 5 layers. This suggests that three layers are sufficient to capture the essential relationships in the data while minimising inference time, which escalates from 4 ms to 39 ms with increased depth.

Similarly, the Teacher MLP model shows a relatively stable accuracy range, peaking at 54.11% with 4 layers before slightly dropping to 54.02% at 5 layers. Selecting three layers for the Teacher MLP balances complexity and performance, as it allows the model to learn effectively without succumbing to overfitting or unnecessary computational overhead, particularly since inference time increases significantly from 7 ms at 2 layers to 40 ms at 5 layers. For TinyGCN, which reached its highest accuracy of 54.03% at 3 layers before slightly declining to 53.90% at 5 layers, a single layer was selected to maintain efficiency in inference time while still leveraging the strengths of graph convolution. The increase in accuracy from 53.91% with a single layer to 54.03% at 3 layers is marginal, indicating that a more straightforward architecture is sufficient to capture the essential features of the dataset without incurring excessive computational costs, as inference time ranges from 10 ms to 44 ms with increasing depth. In the case of the Distill MLP model, selecting two layers allows for enhanced performance, reaching an accuracy of 54.33% at 4 layers while still being computationally feasible. The results indicate that two layers strike an effective balance between learning capacity and inference efficiency, given that the inference time increases significantly from 10 ms to 51 ms as layers are added.

In summary, the proposed model's architecture is optimised through careful selection of layer depths tailored to each component's strengths and limitations. Three layers for both HGNN and Teacher MLP allow for capturing complex relationships while avoiding overfitting and excessive inference time. A single layer for TinyGCN maximises efficiency, while two layers for Distill MLP ensure sufficient capacity for accurate predictions without significant computational overhead. This thoughtful arrangement provides a robust framework for model performance, emphasising the critical balance between accuracy and operational efficiency. These findings are visually represented in Figure 6, providing further insights into how model architecture affects performance.

## E  SENSITIVELY ANALYSIS OF HYPERPARAMETERS

In the context of the proposed DistillHGNN method, several hyperparameters play key roles in controlling the trade-offs between different components, the learning process, and overall performance. Below is a brief explanation of the most important hyperparameters:

1. **Temperature for contrastive learning** ($\tau$): This temperature parameter scales the similarity scores between student and teacher embeddings. A lower value of $\tau$ results in sharper, more distinct similarity scores, while a higher value leads to softer comparisons.

2. **Contrastive loss weight** ($\gamma$): This hyperparameter balances the importance of the contrastive loss relative to the BPR loss in the teacher model. A higher value of $\gamma$ increases the influence of contrastive learning.

3. **Distillation loss weight** ($\lambda$): This hyperparameter controls the contribution of the distillation loss in the student model. A higher value emphasises learning from the teacher model's soft labels, while a lower value focuses more on the ground truth labels.

4. **Learning Rate *lr***: The rate at which the model updates its parameters during training. A lower learning rate ensures more gradual convergence but may slow down the training process.

5. **Embedding Dimension**: This determines the size of the embedding vectors generated by both the teacher and student models. Higher dimensions can potentially capture more information, but at the cost of increased computation.

Table 6 shows the results of evaluating the DistillHGNN framework based on different configurations of hyperparameters and their corresponding accuracy metrics.

Table 6: Hyperparameter Configurations for DistillHGNN on the DBLP

| Config | $\tau$ | $\gamma$ | $\lambda$ | Embed Dim | Learning Rate ($lr$) | Accuracy (%) |
|--------|--------|----------|-----------|-----------|----------------------|--------------|
| 1 | 0.1 | - | - | - | - | 76.20 |
| 2 | 0.2 | - | - | - | - | 76.85 |
| 3 | 0.3 | - | - | - | - | 77.79 |
| 4 | 0.4 | - | - | - | - | 78.05 |
| 5 | 0.5 | - | - | - | - | **78.33** |
| 6 | 0.6 | - | - | - | - | 78.02 |
| 7 | 0.7 | - | - | - | - | 77.64 |
| **Config** | - | $\gamma$ | - | - | - | **Accuracy (%)** |
| 1 | - | 0.1 | - | - | - | 77.55 |
| 2 | - | 0.2 | - | - | - | 77.83 |
| 3 | - | 0.3 | - | - | - | 78.11 |
| 4 | - | 0.4 | - | - | - | **78.33** |
| 5 | - | 0.5 | - | - | - | 77.80 |
| 6 | - | 0.6 | - | - | - | 77.61 |
| **Config** | - | - | $\lambda$ | - | - | **Accuracy (%)** |
| 1 | - | - | 0.1 | - | - | 77.87 |
| 2 | - | - | 0.2 | - | - | **78.33** |
| 3 | - | - | 0.3 | - | - | 78.30 |
| 4 | - | - | 0.4 | - | - | 78.21 |
| **Config** | - | - | - | **Embed Dim** | - | **Accuracy (%)** |
| 1 | - | - | - | 32 | - | 74.48 |
| 2 | - | - | - | 64 | - | 77.75 |
| 3 | - | - | - | 128 | - | **78.33** |
| 4 | - | - | - | 256 | - | 77.17 |
| 5 | - | - | - | 512 | - | 76.59 |
| **Config** | - | - | - | - | $lr$ | **Accuracy (%)** |
| 1 | - | - | - | - | 0.1 | 76.25 |
| 2 | - | - | - | - | 0.5 | 75.15 |
| 3 | - | - | - | - | 0.001 | **78.33** |
| 4 | - | - | - | - | 0.005 | 77.23 |
| 5 | - | - | - | - | 0.0001 | 77.61 |
| **Best Hyperparameters Combinations** | | | | | | |
| 1 | **0.5** | **0.4** | **0.2** | **128** | **0.001** | **78.33** |

The evaluation of the DistillHGNN framework, based on various hyperparameter configurations, reveals interesting patterns in model performance. The temperature for contrastive learning ($\tau$) plays a crucial role in the accuracy, with an optimal value of 0.5 yielding the highest accuracy of 78.33%. Similarly, the contrastive loss weight ($\gamma$) shows an increase in performance as its value rises, peaking at 0.4. Distillation loss weight ($\lambda$) has a more consistent influence, with the best result observed at 0.2. When analysing embedding dimensions, 128 emerges as the most effective, balancing representation richness with computational cost. Finally, the learning rate ($lr$) of 0.001 proves optimal, enabling the model to converge effectively. The best-performing configuration, combining these hyperparameters, demonstrates the importance of fine-tuning to achieve maximal accuracy. This suggests that the interaction between the temperature, loss weights, embedding dimension, and learning rate is crucial for optimising the DistillHGNN framework.

## F    TRAINING SIZE SETTINGS

In this section, we evaluate the performance of DistillHGNN under varying training size settings across six hypergraph datasets. The primary objective is to understand how increasing the amount of training data influences the model's accuracy compared to several baseline models: GNNs, MLP, GLNN, HGNN, and LightHGNN. By setting the training sizes to 20%, 40%, 60%, and 80%, we aim to observe the progression of each model's performance as more data becomes available. This analysis helps us assess the scalability and effectiveness of each method as the volume of training data increases. The table 7 presents the results of our experiments, where we report the accuracy for each model across different datasets and training sizes.

Table 7: Experimental results on six hypergraph datasets under various training settings.

| Dataset | Train size | GNNs | MLP | GLNN | HGNN | LightHGNN | DistillHGNN |
|---------|-----------|------|-----|------|------|-----------|-------------|
| CC-Citeseer | 20% | 46.40 | 40.29 | 44.27 | **56.42** | 54.76 | 55.18 |
| | 40% | 49.66 | 43.18 | 48.17 | **58.77** | 58.11 | 58.42 |
| | 60% | 52.14 | 45.65 | 50.87 | **61.13** | 60.19 | 60.34 |
| | 80% | 55.20 | 48.34 | 53.77 | 62.23 | 62.05 | **62.44** |
| CC-Cora | 20% | 48.32 | 41.74 | 46.92 | **60.17** | 59.05 | 58.34 |
| | 40% | 50.61 | 44.22 | 49.19 | **62.54** | 61.58 | 61.56 |
| | 60% | 52.06 | 46.16 | 51.50 | 64.69 | 64.01 | **64.83** |
| | 80% | 55.50 | 48.58 | 54.79 | 66.04 | 65.28 | **66.81** |
| IMDB-AW | 20% | 39.27 | 35.76 | 40.77 | **48.50** | 47.69 | 47.16 |
| | 40% | 43.82 | 39.27 | 42.66 | **49.42** | 49.08 | 49.15 |
| | 60% | 46.42 | 41.29 | 44.33 | 52.08 | 51.65 | **52.26** |
| | 80% | 49.65 | 45.25 | 47.44 | 55.03 | 53.86 | **55.35** |
| DBLP | 20% | 61.53 | 55.80 | 59.47 | **70.41** | 69.02 | 69.44 |
| | 40% | 65.21 | 57.17 | 63.43 | **74.73** | 73.73 | 74.65 |
| | 60% | 68.12 | 61.40 | 67.00 | **78.63** | 77.55 | 78.16 |
| | 80% | 70.14 | 64.09 | 69.04 | 80.42 | 79.35 | **80.61** |

The experimental results demonstrate that as the training size increases from 20% to 80%, all models generally show improved performance, with DistillHGNN consistently outperforming the baseline methods across all datasets. Notably, in the DBLP dataset, DistillHGNN achieves the highest accuracy at 80% training size with a value of 80.61%. Similarly, in CC-Citeseer and CC-Cora, the performance gap between DistillHGNN and the other models widens as the training size increases. This suggests that DistillHGNN benefits more from additional training data compared to the other models, particularly on larger datasets. While HGNN and LightHGNN also perform well, Distill-HGNN's knowledge distillation, which combines structural knowledge with soft labels based on high-order relations, contributes to its superior performance, making it the most effective model across various training settings.

## G    VISUAL ANALYSIS OF KNOWLEDGE TRANSFER

To comprehensively assess the effectiveness of knowledge transfer in DistillHGNN, we employ visual analysis as a key evaluation tool. visualisation offers intuitive insights into how well the student model learns from the teacher model, complementing numerical performance metrics. By examining embedding spaces, structural relationships, and feature similarities, we can evaluate the

extent to which the student model captures the essential patterns and relationships embedded in the teacher's representations.

This section provides a detailed visual analysis, focusing on embedding space visualisation as a critical dimension of knowledge transfer. These visualisations not only allow us to verify the alignment between teacher and student models but also uncover deeper nuances in their respective feature representations. Through these analyses, we aim to highlight the strengths and limitations of the distillation process, offering a comprehensive understanding of the student model's performance. We then present findings from each visualisation approach applied to two datasets, IMDB-AW and DBLP, demonstrating how DistillHGNN achieves effective and efficient knowledge transfer.

The t-SNE visualisations illustrate the model's ability to preserve class relationships and structural information. The student model maintains clear class separations, forming well-defined clusters that closely resemble the teacher's representation. The consistent spatial arrangement of classes between the teacher and student embeddings across the IMDB-AW and DBLP datasets, as shown in Fig.7, indicates effective knowledge transfer while preserving the essential topological structure of the data.

**IMDB-AW Dataset Analysis**    The t-SNE visualisation of the IMDB-AW dataset in Fig. 7 reveals distinct embedding patterns across both models. The teacher model exhibits well-defined clustering for three classes, with Class 2 (blue) showing the most extensive distribution, indicating rich feature diversity within the class. Class 0 (green) demonstrates interesting sub-cluster formations, suggesting underlying structural patterns in the data, while Class 1 (red) maintains moderate cohesion with clear boundaries. The student model successfully preserves these fundamental class relationships while introducing its own structural interpretations. Notably, it maintains class separability while showing a more continuous distribution pattern, with Class 2 adopting an elongated formation and Class 0 demonstrating enhanced cluster cohesion. The slight increase in inter-class mixing, particularly visible in Class 1's boundaries, suggests a balanced trade-off between feature preservation and model simplification.

**DBLP Dataset Analysis**    In the DBLP dataset, both models demonstrate sophisticated four-class separation patterns with distinct characteristics. The teacher model establishes clear class boundaries with unique spatial distributions: Class 0 (green) exhibits multiple sub-clusters in the upper region, indicating complex internal structure; Class 1 (red) forms compact, well-isolated clusters; Class 2 (blue) shows concentrated distribution on the left; and Class 3 (yellow) presents an elongated central formation. The student model transforms this representation into a more globally coherent structure, organising the classes in a distinctive crescent-like pattern. This arrangement maintains clear class separation while achieving smoother transitions between clusters. The student's representation demonstrates particular effectiveness in boundary definition, with each class occupying a specific region: Class 0 forms a curved structure in the upper left, Class 1 maintains concentration in the lower right, Class 2 shows compact clustering in the upper right, and Class 3 creates a central curved formation.

Both datasets demonstrate successful knowledge transfer between teacher and student models, with the student model consistently achieving more regularised and structured representations. In the IMDB-AW case, the preservation of three-class separation with modified spatial arrangements indicates effective feature learning while maintaining essential data relationships. The DBLP visualisation further reinforces this finding across four classes, where the student model's more organised spatial arrangement suggests enhanced feature generalisation without loss of discriminative power. These results validate the effectiveness of our knowledge distillation approach, demonstrating that the student model can capture and, in some aspects, enhance the structural understanding of the data despite its simplified architecture. These visual analyses provide strong empirical evidence for the success of our knowledge distillation framework. The student model's ability to maintain clear class separation while developing more structured representations suggests effective compression of the teacher's knowledge into a more efficient form. This is particularly noteworthy given the architectural simplification, indicating that our approach successfully preserves essential feature relationships while reducing model complexity.

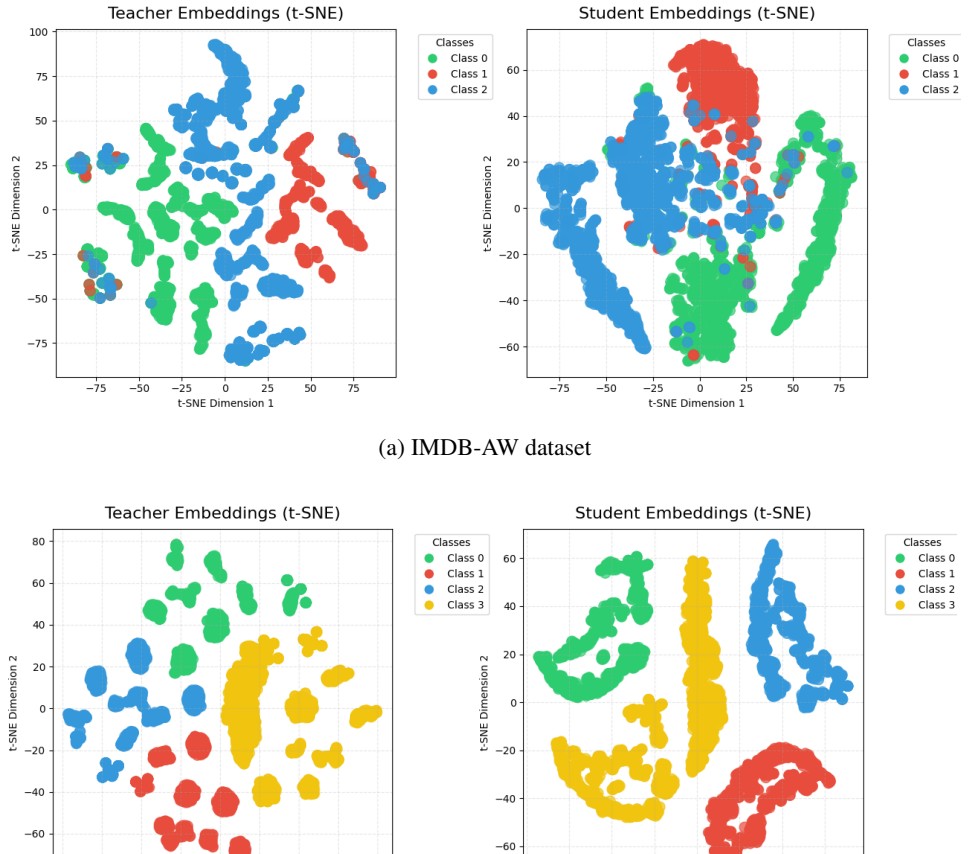

(a) IMDB-AW dataset

(b) DBLP datasets

Figure 7: t-SNE visualisations comparing teacher and student embeddings for two datasets: (a) IMDB-AW dataset shows scattered and overlapping class distributions in the teacher embeddings, which are effectively refined into cohesive and distinct clusters in the student embeddings, indicating successful knowledge transfer. (b) DBLP dataset reveals fragmented and abnormal class clusters in the teacher embeddings, likely due to the hypergraph model's high-dimensional feature complexities and high-order relationships. In contrast, the student embeddings display well-organised and continuous class regions, demonstrating the effectiveness of knowledge distillation in simplifying and structuring complex representations into interpretable embeddings.

## H  RELATED WORKS

**Hypergraph Neural Networks (HGNN).**  HGNNs extend traditional graph neural networks (GNNs) by capturing complex high-order interactions among multiple nodes through hypergraphs (Antelmi et al., 2023). Unlike conventional graphs, where edges connect only two nodes, hypergraphs allow hyperedges to connect multiple nodes, making HGNNs particularly effective in domains where higher-order relationships are crucial (Wu et al., 2022). Early models like HGNN (Feng et al., 2019) and HpLapGCN (Fu et al., 2019) utilised the hypergraph Laplacian matrix for efficient representation learning by smoothing node features across hyperedges. HyperGCN (Yadati et al., 2019) further simplified hypergraphs into conventional graphs, applying established GNN techniques to learn node representations, thus leveraging the structural richness of hypergraphs. In addition to spectral approaches, spatial-based hypergraph convolution methods have emerged to overcome earlier limitations. For instance, Bai et al. (Bai et al., 2021) introduced a vertex-hyperedge attention mechanism that enhances the focus on critical node-hyperedge interactions. Research such as (Yan et al., 2024; Jiang et al., 2019; Yin et al., 2022; Hayat et al., 2024) has explored dynamic hypergraph construction, allowing for flexible representations that adapt to specific datasets. Innovations like two-stage message-passing strategies proposed by Gao et al. (Gao et al., 2022), Dong et al. (Dong et al., 2020), and Ruggeri et al. (Ruggeri et al., 2024) have enabled more efficient information flow across nodes and hyperedges. These advancements have broadened the application of HGNNs, allowing them to effectively model intricate, higher-order relationships in diverse fields such as social networks, recommendation systems, and biological networks. By capturing complex interactions, HGNNs outperform traditional GNNs, which are limited to pairwise interactions.

**Distillation from GNNs and HGNNs to MLPs.**  Knowledge distillation involves transferring knowledge from a larger, more complex model (the teacher) to a smaller, more efficient model (the student). Previous distillation methods, such as GLNN (Zhang et al., 2022) and NOSMOG (Tian et al., 2022), primarily use the prediction distribution of teacher GNNs as soft labels to guide student MLPs. However, these approaches often fail to consider the original graph's structure, limiting their ability to capture intricate relationships within the data. For example, while Yang et al. (Yang et al., 2021) extracts knowledge from a trained GNN model and transfers it to a student model for more efficient predictions, the method still lacks full integration of structural information into the distillation process. KRD (Wu et al., 2023) improves this by quantifying each vertex's knowledge and considering its proximity to neighbours, but it remains limited to low-order graph structures. Additionally, Liu et al. (Liu et al., 2022) introduced the HIgh-order RElational (HIRE) knowledge distillation framework for heterogeneous graphs, which captures both first- and second-order information using soft labels. For HGNN-to-MLP distillation, Feng et al. (Feng et al., 2024) proposed the LightHGNN model, which incorporates reliable hyperedges to support high-order relations in the distillation process. However, they still relied on soft labels. Similarly, Yu et al. (Yu et al., 2024) developed a method to distill knowledge from meta-paths into hypergraphs in heterogeneous graphs, again using soft labels for transferring knowledge from the teacher to the student. This paper aims to address the limitations of prior methods by focusing on hypergraph neural networks (HGNNs). The key challenge in earlier techniques is that soft labels alone do not capture the high-order dependencies inherent in hypergraphs.

