# OpenReview forum: "DistillHGNN: A Knowledge Distillation Approach for High-Speed Hypergraph Neural Networks"
_ICLR.cc/2025/Conference — ICLR 2025 Poster_

### Official Review · Reviewer_49KC · 2024-11-01

**Soundness:** 2
**Presentation:** 2
**Contribution:** 2
**Rating:** 5
**Confidence:** 5

**Summary:**

This paper proposes **DistillHGNN**, a knowledge distillation framework designed to accelerate the inference of hypergraph neural networks (HGNNs) while maintaining their high accuracy. DistillHGNN leverages a teacher-student structure where the teacher model, an HGNN, captures complex relationships within hypergraphs and generates soft labels. The student model, a simplified TinyGCN, learns from these soft labels and employs contrastive learning to capture high-order structural knowledge.

**Strengths:**

1. The dual use of soft labels and structural knowledge distillation through contrastive learning enhances the capability of the student model, addressing limitations of previous distillation methods.
2. DistillHGNN achieves significant reductions in inference time (up to 80%) compared to traditional HGNNs, making it suitable for real-time applications.

**Weaknesses:**

1. **(Minor)** The writing could be improved for clarity and simplicity. For instance, the caption in Figure 1 (lines 175–188) could be condensed to improve readability, ideally to half its current length. If extensive detail is necessary, consider moving some information to the main text.
2. **(Minor)** The paper could use `\citet{}` for citations when referring to the authors directly. For example, in lines 280–281, instead of "Multilayer Perceptron (MLP) by \cite{author2020}," using `\citet{}` could enhance readability.
3. When mentioning the use of six popular datasets, it would be helpful to list the features and characteristics of each dataset to provide more context for the experiments. For instance, some datasets may have dense connections while others are sparse; some may benefit more from heterogeneous graphs compared to homogeneous ones. Without a detailed explanation of the datasets' biases or inductive properties, understanding the generalization of the proposed method becomes challenging.
4. **Scalability**: The scalability of the proposed method is unclear. According to Table 3, all datasets used in the experiments are relatively small, with node counts between 3k and 6k, similar in scale to the Cora dataset. It would be beneficial to provide insights on how the method scales to larger datasets, such as ogbn-arxiv or ogbn-products, if they have hypergraph structures.

**Questions:**

See Weaknesses part

---

> ### Author Response · Authors · 2024-11-28
> **We have shortened the caption of Figure 1 for clarity, revised author references to use \citet{}, and expanded Table 3 to include key structural properties of eight datasets. We also added IMDB and DBLP datasets to evaluate scalability (Table 1), and clarified that while larger datasets like ogbn-arxiv were not used due to resource constraints, future work will address this.**
>
> We sincerely appreciate your detailed and thoughtful feedback, which has significantly contributed to enhancing the quality of our manuscript.
>
> Weaknesses:
>
> 1) We agree that the caption could be more concise while retaining clarity. In the revised version, we have shortened the caption of Figure 1 to highlight the essential elements.
>
>
> 2) We have revised all direct author references throughout the paper to use citet{} instead of cite{}.
>
>
> 3) We have expanded Table 3 to include key structural properties of the six datasets, providing context for the experiments:
>
> D1) CC-Citeseer: Sparse, homogeneous citation network (3,312 nodes, 1,004 hyperedges, avg. degree 3.2), testing robustness to sparse, biased connections.
>
> D2) CC-Cora: Moderately sparse, homogeneous network (2,708 nodes, 1,483 hyperedges, avg. degree 3.8), highlighting modular structures.
>
> D3) IMDB-AW: Dense, heterogeneous actor-movie network (5,355 nodes, 6,811 hyperedges, avg. degree 8.4), suitable for testing inductive learning in dynamic scenarios.
>
> D4) DBLP-paper: Heterogeneous paper-author network (14,376 nodes, 14,475 hyperedges, avg. degree 5.2), focusing on collaborative dynamics.
>
> D5) DBLP-term: Dense, heterogeneous term-paper network (14,376 nodes, 13,789 hyperedges, avg. degree 7.1), capturing semantic clustering.
>
> D6) DBLP-Conf: Sparse, heterogeneous conference-paper network (14,376 nodes, 1,612 hyperedges, avg. hyperedge size 284.2), emphasizing hierarchical organisation.
>
> D7) DBLP: Large, dense, heterogeneous network (66,543 nodes, 274,824 hyperedges, avg. degree 8.26), exploring collaborative and citation dynamics.
>
> D8) IMDB: Very dense, heterogeneous network (142,129 nodes, 1,596,148 hyperedges, avg. degree 22.46), validating scalability on highly connected, complex structures.
>
> These datasets span diverse densities, heterogeneity, and structures, validating our method's ability to generalise across varying graph characteristics effectively.
>
>
> 4) Thank you for your insightful comment. The datasets we used in our experiments were selected to ensure comparability with prior works and baseline models, which also utilized these datasets. To address your concern regarding scalability, we extended our evaluations to include two large heterogeneous datasets: IMDB (142,129 nodes, 1.59M edges) and DBLP (66,543 nodes, 274,824 edges). These results, based on accuracy metrics, are included in the revised manuscript and in Table 1.
>
> While larger datasets like ogbn-arxiv (169K nodes) and ogbn-products (2.4M nodes) would provide further insights, our evaluations demonstrate that DistillHGNN scales efficiently to datasets of comparable complexity, such as IMDB, which is of a similar size to ogbn-arxiv.
> Due to GPU resource limitations and time constraints, we were unable to evaluate on these larger datasets in this submission. However, we plan to include these experiments in future work to further validate the scalability of DistillHGNN.

---

### Official Review · Reviewer_ErwD · 2024-11-02

**Soundness:** 3
**Presentation:** 3
**Contribution:** 2
**Rating:** 6
**Confidence:** 5

**Summary:**

This paper introduces DistillHGNN, a novel knowledge distillation framework designed to enhance the inference speed and memory efficiency of Hypergraph Neural Networks (HGNNs) while maintaining high accuracy. DistillHGNN employs a teacher-student model where the teacher consists of an HGNN and a Multi-Layer Perceptron (MLP) that generates soft labels. The student model uses a lightweight Graph Convolutional Network (TinyGCN) paired with an MLP, optimized for faster online predictions. Additionally, contrastive learning is utilized to transfer high-order and structural knowledge from the HGNN to the TinyGCN. Experimental results on various real-world datasets demonstrate that DistillHGNN significantly reduces inference time and achieves accuracy comparable to or better than state-of-the-art methods like LightHGNN.

**Strengths:**

1. The proposed method combines knowledge distillation with contrastive learning in an innovative way, allowing for a more comprehensive transfer of both soft labels and high-order structural knowledge from the complex HGNN to the lightweight TinyGCN. This dual transfer mechanism enhances the student model's ability to capture intricate dependencies while maintaining computational efficiency.
2. The experimental evaluation is thorough and spans multiple real-world datasets, including CC-Citeseer, CC-Cora, IMDB-AW, and various DBLP datasets. The results consistently demonstrate that DistillHGNN achieves a favorable balance between accuracy and inference speed, often outperforming existing methods such as LightHGNN in both metrics.
3. The paper is well-structured and provides a clear and detailed explanation of the methodology, including the architecture of both the teacher and student models, the training process, and the loss functions used. This clarity facilitates understanding and potential replication of the proposed framework.

**Weaknesses:**

1. The method involves converting a hypergraph to a homogeneous graph, which can lead to the loss of high-order structural information inherent in hypergraphs. While contrastive learning helps mitigate this issue, the paper does not sufficiently address how the conversion process preserves the complex relationships that hypergraphs naturally capture.
2. DistillHGNN relies on homogeneous graphs during the inference phase, which poses a significant limitation for inductive learning scenarios. When new samples are introduced, the need to reconstruct the graph and perform complex preprocessing steps can result in increased computational and time costs, limiting the model's applicability in dynamic or real-time environments.
3. The evaluation is primarily conducted on datasets related to academic papers and movie databases. There is a lack of assessment on larger-scale or synthetic datasets, which raises concerns about the method’s scalability and generalizability to other domains or more complex real-world applications.
4. The selection of baseline methods, while covering traditional GNNs and some knowledge distillation approaches, does not include more recent or advanced hypergraph neural network models. This omission may limit the understanding of DistillHGNN's relative performance against the latest state-of-the-art techniques.

**Questions:**

1. The proposed method converts a hypergraph into a homogeneous graph, potentially leading to the loss of high-order structural information. How do you address this information loss, and can you incorporate mechanisms similar to LightHGNN’s high-order soft-target constraints to better preserve high-order knowledge? Additionally, could you provide visualizations to enhance the interpretability of your method?
2. Since DistillHGNN depends on homogeneous graphs during inference, how does it handle inductive learning scenarios where new samples are added? Specifically, what strategies do you propose to minimize the need for graph reconstruction and complex preprocessing steps in such cases?
3. The experimental evaluation primarily focuses on datasets from academic papers and movie databases. Do you have plans to evaluate DistillHGNN on larger-scale or synthetic datasets to demonstrate its scalability and effectiveness across a broader range of applications?
4. Could you provide more technical details and visualizations regarding how contrastive learning and soft labels facilitate the transfer of high-order knowledge from the teacher model to the student model? This would help in better understanding the mechanisms behind knowledge preservation and transfer.

---

> ### Author Response · Authors · 2024-11-28
> **Our approach extends LightHGNN with a dual knowledge transfer mechanism for better high-order information preservation, utilizing both structural and predictive knowledge transfer via contrastive learning and soft labels. The simplified architecture of TinyGCN enables efficient processing of new nodes without graph reconstruction.**
>
> Thank you for your comprehensive and valuable comments, which have played a crucial role in improving the manuscript.
>
> Weaknesses:
>
> 1) Our approach advances beyond LightHGNN through a dual knowledge transfer mechanism that ensures comprehensive preservation of high-order information:
>
> A. Structural Knowledge Transfer:
> - Teacher HGNN captures hypergraph structure through the hypergraph Laplacian (Eq. 1)
> - This structural knowledge is transferred via contrastive learning using InfoNCE loss (Eq. 6), which ensures alignment between teacher embeddings $Z^t$ and student embeddings $Z^s$
>
> B. Predictive Knowledge Transfer:
> - Soft labels $Y_v^t$ from the teacher model capture class probability distributions.
> - Knowledge is transferred through KL divergence in the student loss (Eq. 12), preserving the teacher's prediction patterns.
>
> Unlike LightHGNN which relies solely on soft-target constraints, our method provides:
> i) Direct embedding alignment through contrastive learning
> ii) Explicit preservation of hypergraph structure through InfoNCE loss
> iii) Fine-grained knowledge transfer via both embeddings and probability distributions
>
> 2) The student model (TinyGCN) utilizes a simplified architecture that operates on node features (X) and a basic adjacency matrix $(A^s)$, enabling efficient processing of new nodes without full graph reconstruction. The single-layer aggregation $(Z^s = A^s X W^s)$ reduces computational overhead and allows incremental processing using local neighborhood information.
>
> Additionally, the student model captures high-order relationships through the teacher’s guidance, with the learned weight matrix (W^s) facilitating efficient application to new nodes without relying on the original hypergraph structure.
>
> Thus, our model's simplified design and efficient knowledge transfer make it suitable for dynamic environments, addressing inductive learning challenges while maintaining performance.
>
> 3) The datasets used in our experiments were selected to align with prior works and baseline models. We have added the IMDB dataset (142,129 nodes, 1.59M edges) and the DBLP dataset (66,543 nodes, 274,824 edges) to Table 1, as shown in the revised version.
> 4) Our analysis includes models such as LightHGNN (Feng et al., 2024) and GLNN (Zhang et al., 2021). To provide a more comprehensive evaluation, we have expanded the comparison in the revised version to include the seminal GCN model (Kipf & Welling, ICLR 2017) and KRD (Wu, ICML 2023) in Table 1.
>
> Questions:
>
> 1) Please refer to Weakness 1.
> 2) Please refer to Weakness 2.
> 3) Please refer to Weakness 3.
> 4) We added the section "Visual Analysis of Knowledge Transfer" and included t-SNE visualizations comparing teacher and student embeddings in Figure 7 in the revised version of the manuscript. On the other hand, our dual-channel knowledge preservation operates as follows:
>
> 1. High-Order Knowledge Transfer via Message Passing and Contrastive Learning
>
> - The teacher HGNN employs message passing through multiple hypergraph convolution layers ($H^1 \rightarrow H^2 \rightarrow \cdots \rightarrow H^L$), where each layer aggregates information from higher-order neighborhoods using the hypergraph incidence matrix $\mathcal{H}$.
>  - TinyGCN preserves this structural information through simplified message passing on the homogeneous graph ($A = \mathcal{H}W\mathcal{H}^T - D_v$).
>  - The contrastive loss ($\mathcal{L}_{\text{Con}}$) ensures the alignment between teacher embeddings $Z^t$ and student embeddings $Z^s$, effectively transferring the message-passing patterns.
>
> 2. Knowledge Preservation through Soft Labels
>  - The teacher's MLP generates soft labels ($Y^t$) that encode the high-order patterns captured through iterative message passing in HGNN layers.
> - These soft labels are distilled into the student using the distillation component of $\mathcal{L}_{\text{student}}$.
> - This process ensures the student captures both local and global structural information, combining supervised learning and distillation.
>
> Together, these mechanisms complement each other: message passing and contrastive learning preserve structural relationships in the embedding space, while soft label distillation ensures the transfer of high-order predictive patterns. Figure 1 visually illustrates how these components work in unison.

---

> > ### Comment · Reviewer_ErwD · 2024-12-02
> >
> > The author has addressed most of my concerns. This paper focuses on achieving a balance between HGNN and LightHGNN, improving speed compared to HGNN and enhancing accuracy compared to LightHGNN. I have decided to increase the score for this paper. Good luck!

---

### Official Review · Reviewer_6QuE · 2024-11-02

**Soundness:** 2
**Presentation:** 3
**Contribution:** 3
**Rating:** 8
**Confidence:** 4

**Summary:**

This paper proposed DistillHGNN, a novel framework designed to improve the inference speed of Hypergraph Neural Networks (HGNNs) without sacrificing accuracy. DistillHGNN utilizes a teacher-student knowledge distillation approach, where the teacher model comprises an HGNN and a Multi-Layer Perceptron (MLP), while the student model, TinyGCN, is a lightweight Graph Convolutional Network (GCN) paired with an MLP. The framework incorporates a dual knowledge transfer mechanism, passing both soft labels and structural knowledge from the teacher to the student. Trained on labeled data and teacher-provided soft labels with contrastive learning to retain high-order relationships, DistillHGNN achieves performance comparable to traditional HGNNs with significantly lower resource requirements.

**Strengths:**

- The dual knowledge distillation approach (soft labels and high-order structural knowledge) effectively transfers complex relationships, addressing limitations found in existing methods.

- Comparative accuracy with minimal inference time.

- The paper is well structured and well written.

- The paper includes a thorough experimental setup, covering multiple datasets and evaluation metrics.

**Weaknesses:**

- The authors claim the proposed approach is memory efficient everywhere in the paper. In line 416, they mention, "As illustrated in Figure 2, DistillHGNN maintains competitive inference times and memory efficiency without sacrificing accuracy"; however, Figure 2 shows accuracy vs. inference time. Moreover, no other evidence shows that the proposed approach is memory efficient.

- Among the previous works in knowledge distillation, the authors only compared with LightHGNN. Since the authors compared with GNN and GCN, the authors are suggested to compare with GNN and GCN-based distillation approaches as well.


Minor:

- In line 152, the authors mention "Framework 1". Since there is no Framework 1 in the paper, I suppose it should be "Figure 1".

- The caption for Figure 1 is too big. The authors should keep it concise and incorporate the details into the actual text of Section 3. Also, the authors didn't mention what happens with the soft labels after they are generated from the teacher model in the caption.

**Questions:**

- Figure 2 shows comparisons of models based on accuracy and inference time. Does that mean accuracy and inference time are dependent on each other? If not, these graphs create confusion. Currently, from the graphs it seems that if the inference time is longer, the accuracy could be higher, and vice versa. If this is not the case, I suggest keeping inference time separate from accuracy.

- Line 669 of Algorithm 1 states, "Update teacher model parameters via backpropagation". Typically, knowledge distillation keeps the teacher fixed once pre-trained, so if this algorithm intends otherwise, the authors should provide a justification.

- In Figure 6, can the authors keep the range of accuracy axis the same for all the graphs? It would be easier to visualize the differences.

- It would be interesting to see how an adaptive knowledge distillation approach performs compared to the proposed fixed distillation approach. For instance, the model could prioritize contrastive learning for highly connected hypergraphs to capture complex relationships and rely more on soft labels for simpler ones to reduce model complexity even further.

---

> ### Author Response · Authors · 2024-11-28
> **We have carefully addressed the comments and suggestions provided by the reviewer. Key points include clarifications regarding memory efficiency based on the architecture of the student model, enhancements to the comparative evaluation with baseline models, and corrections in Figure 1 and related sections.**
>
> Thank you for the thorough and insightful comments, which have greatly helped us improve the manuscript.
>
> Weaknesses:
>
> 1)  You are absolutely right. Our claim regarding memory efficiency is primarily based on the architectural design of our student model, which utilizes a simplified TinyGCN with a single layer. This streamlined architecture inherently reduces memory consumption by efficiently distilling high-order relational structures from the teacher model into a more compact representation. We have revised this sentence in the updated version.
>
> 2) In terms of comparative evaluation with existing distillation approaches, we have included multiple baseline comparisons. Our analysis encompasses models such as GLNN (GNN-to-MLP), as detailed in Table 1, and KRD (GNN-to-MLP), as presented in Table 2. To ensure a more comprehensive evaluation, we have expanded the comparison in the revised version to include the GCN model (Kipf & Welling, ICLR 2017) and KRD method (Wu, ICML 2023) in the updated Table 1.
>
> Minor:
>
> 1) You are correct; "Framework 1" was intended to refer to "Figure 1." We appreciate your attention to detail and have made the necessary correction in the revised version. Thank you for pointing this out.
>
> 2) We have revised the caption for Figure 1 to be more concise and moved additional details to Section 3, as suggested. The updated caption now includes the role of soft labels in the proposed approach. Specifically, the teacher model (HGNN) generates two key outputs: (1) node embeddings $ Z^t $, derived from the hypergraph Laplacian (Eq. 5), which capture high-order structural relationships, and (2) soft labels $ Y_v^t $, which encode these high-order relations and are produced through the HGNN-MLP pipeline.
>
> These soft labels serve a dual purpose: first, they are incorporated into the teacher model’s supervised loss function (as the second term) to ensure alignment between the teacher’s predictions and the ground truth when available. Second, in cases of insufficient labeled data, these soft labels function as pseudo-labels, facilitating effective training of the student model.
>
> Questions:
>
> 1)  Generally, higher accuracy in models, such as traditional GNNs and GCNs with fully connected layers, often requires increased inference time. However, in Figure 2, we compare models based on both accuracy and inference time. This comparison does not imply a direct dependency between the two. Instead, DistillHGNN and LightHGNN, strives to balance accuracy and inference time effectively. The goal of knowledge distillation models, such as ours, is to achieve this balance while improving both metrics simultaneously.
>
> 2) We appreciate your insightful observation and agree with your concern regarding the teacher model's parameter updates. To clarify, as illustrated in Figure 1, the teacher model undergoes an independent pre-training phase using the supervised loss defined in Eq. 10. During this phase, the teacher’s parameters, which include those for the HGNN and the MLP, are updated to ensure the model captures the high-order structural relationships inherent in the hypergraph.
>
> After this pre-training phase, the teacher model’s parameters are frozen. The student model is then trained separately using its own loss function, with coordination between the teacher and student achieved via the distillation of embeddings and soft labels. This ensures the student learns from the high-order relationships distilled from the teacher while preserving the teacher’s pre-trained knowledge. To address the inconsistency noted in Algorithm 1, we have revised the pseudocode to explicitly reflect this process.
>
> 3) In the revised version, we standardized the accuracy axis range in Figure 6 to enhance clarity and facilitate comparison.
>
> 4) Thank you for the thoughtful suggestion regarding adaptive knowledge distillation. While our fixed distillation approach effectively balances performance and complexity, we acknowledge the potential of an adaptive mechanism. We opted for a fixed approach because:
>
>     A. It ensures stable knowledge transfer between HGNN (teacher) and TinyGCN (student).
>     B. Manual tuning of hyperparameters ($\gamma$, $\lambda$) provides dataset-specific adaptation.
>     C. Its simplicity aligns with our goal of reducing computational complexity.

---

> > ### Comment · Reviewer_6QuE · 2024-11-29
> >
> > Thank you for your detailed response to my concerns.
> >
> > I am still not convinced about the memory efficiency. The paper claims that the proposed method, LightHGNN, enhances memory efficiency, which necessitates a quantitative comparison. Given that LightHGNN and other state-of-the-art methods like GLNN and KRD utilize only a distilled MLP during inference, a configuration that should inherently be memory efficient, it is essential to present empirical data comparing these models with the proposed one to substantiate such claims.

---

> > > ### Author Response · Authors · 2024-11-30
> > > **Memory Efficiency Analysis**
> > >
> > > Thank you for your valuable comment. We want to clarify that we did not claim in the manuscript that our proposed model has better inference time than other distillation models such as LightHGNN, GLNN, or KRD. Instead, we claim that DistillHGNN is more memory-efficient than traditional GNN and HGNN models. To increase the clarity of our claim, we provide a detailed analysis of DistillHGNN’s memory characteristics below:
> > >
> > > Our claim regarding memory efficiency is based on the architectural design of DistillHGNN, which employs a simplified TinyGCN with a single layer. To quantify this, we calculate the memory requirements for each model under typical settings, based on real-world scenarios such as social networks (e.g., Twitter), as analyzed in [1].
> > >
> > > For a GNN with $L$ layers, $R$ neighbors, and $d$-dimensional embeddings, the memory requirement is $\mathcal{O}(R^L \times d)$ due to the need to fetch neighbors. In contrast, DistillHGNN reduces this to $\mathcal{O}(R \times d + d)$, while GLNN/LightHGNN requires only $\mathcal{O}(d)$. Using typical values ($L={3,4,5}$, $R=208$, $d=128$), the memory consumption is as follows:
> > >
> > > 1. Traditional GNN (3, 4, 5 layers):
> > >
> > > For 3 layers:
> > >
> > > $\text{Memory}_{\text{GNN}} = R^L \times d = 208^3 \times 128$
> > >
> > > $\text{Memory}_{\text{GNN}} = 5,674,240 \, \text{units} \approx 5.67 \, \text{MB}.$
> > >
> > > For 4 layers:
> > >
> > > $\text{Memory}_{\text{GNN}} = R^L \times d = 208^4 \times 128$
> > >
> > > $\text{Memory}_{\text{GNN}} = 1,180,161,280 \, \text{units} \approx 1,180.16 \, \text{MB}$.
> > >
> > > For 5 layers:
> > >
> > > $\text{Memory}_{\text{GNN}} = R^L \times d = 208^5 \times 128$
> > >
> > > $\text{Memory}_{\text{GNN}} = 245,736,238,080 \, \text{units} \approx 245,736.24 \, \text{MB}$.
> > >
> > >
> > >
> > > 2. DistillHGNN (Single-layer TinyGCN):
> > >
> > > $\text{Memory}_{\text{DistillHGNN}} = R \times d + d = 208 \times 128 + 128$.
> > >
> > > $\text{Memory}_{\text{DistillHGNN}} = 26,752 \, \text{units} \approx 0.0268 \, \text{MB}$.
> > >
> > >
> > >
> > > 3. GLNN/LightHGNN (MLP):
> > >
> > > $\text{Memory}_{\text{GLNN/LightHGNN}} = d = 128 \, \text{units}$.
> > >
> > > $\text{Memory}_{\text{GLNN/LightHGNN}} = 128 \, \text{units} \approx 0.000128 \, \text{MB}$.
> > >
> > > The memory usage of traditional GNNs grows exponentially with the number of layers. For example:
> > >
> > > - 3 layers: $\sim5.67$ MB
> > >
> > > - 4 layers: $\sim1,180.16$ MB
> > >
> > > - 5 layers: $\sim245,736.24$ MB
> > >
> > > This rapid growth highlights the impracticality of using deep GNNs in memory-constrained environments. In contrast, due to its simplified, single-layer architecture, DistillHGNN maintains a low memory footprint of $\sim0.0268$ MB, regardless of the number of layers.
> > >
> > > GLNN/LightHGNN, which uses an MLP-based inference design, consumes the least memory at $\sim0.000128$ MB. However, this minimal memory usage comes at the cost of reduced accuracy. These models cannot capture or transfer structural knowledge, essential for optimizing performance in more complex tasks.
> > >
> > > DistillHGNN overcomes this limitation by introducing a dual knowledge transfer mechanism that uses soft labels and structural information. This enables DistillHGNN to significantly outperform GLNN and LightHGNN in terms of accuracy (as shown in Table 1), while maintaining a modest memory usage of $\sim0.0268$ MB. The added memory overhead is negligible in real-world computational settings, especially when balanced against the substantial accuracy gains.
> > >
> > > Moreover, the exponential memory growth in traditional GNNs (e.g., reaching $\sim245$ GB for 5 layers) starkly contrasts with DistillHGNN's efficiency. Although DistillHGNN has a slightly higher inference time compared to GLNN/LightHGNN, this is a deliberate design choice aimed at prioritizing accuracy, while still ensuring computational feasibility. This balance between accuracy and efficiency establishes DistillHGNN as a robust solution, effectively addressing the limitations of existing methods without compromising resource efficiency.
> > >
> > > To address your concern, we will perform the experiments again to report the exact memory consumption of DistillHGNN and compare it with that of the baseline models in the final version of the manuscript.
> > >
> > >
> > > [1] Zhang, S., Liu, Y., Sun, Y., \& Shah, N. (2022). Graph-less neural networks: Teaching old MLPs new tricks via distillation. \textit{ICLR 2022}.

---

> > > > ### Comment · Reviewer_6QuE · 2024-12-02
> > > >
> > > > I highly recommend that the authors incorporate a memory versus accuracy trade-off analysis into the paper. This addition would address an essential aspect of the research. Aside from this, most of my concerns have been addressed. I have decided to increase the score for this paper. Good luck!

---

### Meta-Review · Area_Chair_zCdB · 2024-12-22

**Metareview:**

This paper introduces DistillHGNN, a knowledge distillation framework designed to improve the efficiency of Hypergraph Neural Networks (HGNNs). It employs a teacher-student paradigm where the teacher is a combination of HGNN and MLP. The framework aims to reduce inference time and memory usage while preserving high accuracy through dual knowledge transfer, by using soft labels and contrastive learning. Experimental results confirm the effectiveness of the proposed method.

The overall quality of this paper is good. It is well-written and well-motivated. The proposed method is new and reasonable. The experimental results are extensive and can support the proposed method.

**Additional Comments On Reviewer Discussion:**

This paper finally receives the scores of 8 (Reviewer 6QuE), 6 (Reviewer ErwD), 5 (Reviewer 49KC). The main concern of Reviewer 49KC lies in that the datasets used in the experiments are relatively small. The authors have provided a rebuttal and incorporated two more large datasets (IMDB and DBLP). Reviewer 49KC did not provide a response to the authors' rebuttal, but I feel that the main concern was addressed as two more large datasets are used.

Considering that the average score of this paper is relatively high and I feel that the overall quality of this paper is good. I recommend accepting this paper. I suggest the authors could consider making necessary changes as indicated by the reviewers, e.g., experimental comparisons with GNN and GCN based distillation approaches, providing visualizations to enhance the interpretability, more experiments on large-scale datasets, etc.

---

### Decision · Program_Chairs · 2025-01-22

Accept (Poster)